# Geoelectrochemistry-driven alteration of amino acids to derivative organics in carbonaceous chondrite parent bodies

Yamei Li [1], Norio Kitadai [1,2], Yasuhito Sekine [1,3], Hiroyuki Kurokawa [1], Yuko Nakano[1] & Kristin Johnson-Finn[1,4]

A long-standing question regarding carbonaceous chondrites (CCs) is how the CCs' organics were sourced and converted before and after the accretion of their parent bodies. Growing evidence shows that amino acid abundances in CCs decrease with an elongated aqueous alteration. However, the underlying chemical processes are unclear. If CCs' parent bodies were water-rock differentiated, pH and redox gradients can drive electrochemical reactions by using $H_2$ as an electron source. Here, we simulate such redox conditions and demonstrate that $\alpha$-amino acids are electrochemically altered to monoamines and $\alpha$-hydroxy acids on FeS and NiS catalysts at 25 °C. This conversion is consistent with their enrichment compared to amino acid analogs in heavily altered CCs. Our results thus suggest that $H_2$ can be an important driver for organic evolution in water-rock differentiated CC parent bodies as well as the Solar System icy bodies that might possess similar pH and redox gradients.

Organics in carbonaceous chondrites (CCs) and other extra-terrestrial bodies record a mixed history of solar system chemistry and parent body processes. Amino acids of many isomeric and homolog varieties have been repeatedly observed in CCs and invoked as key prebiotic molecules involved in the origin of life on Earth and possibly elsewhere[1,2]. In the past decade, the advancement of analytical instrumentations, an increasing supply of samples from various CC subgroups, and the development of a more accurate petrographic classification system of CCs (Harju/Rubin scale)[3,4] have facilitated the study on how parent body processes have shaped the organic distributions. However, there is a large discrepancy between experimental results and meteoritic records. Many aqueous syntheses of amino acids have been demonstrated through either Strecker-[5,6] or Formose-type[7], and reductive amination[8,9] reactions, which suggest aqueous alteration would promote the accumulation of amino acids. Nevertheless, meteoritic records have shown that heavily aqueously altered CCs showed

markedly lower total amino acid abundances than more primitive ones[10–15]. Recently, such a tendency was further confirmed in separate sets of CM[14] and CR chondrites[13], pointing to a potential "water paradox". The "waterparadox" is mainly due to the unknown processes of amino acid alteration in aqueous solution, especially at low temperatures[16,17]. Although being invoked as a driver for amino acid alteration, hydrothermal decomposition of amino acids occurs at temperatures higher than 200 °C[17–21], which is much higher than the temperatures these CCs have experienced (e.g., CM chondrites, 0~25 °C; CR chondrites, ≤150 °C; CI chondrites, 50~150 °C)[22,23]. Photochemistry can decompose amino acids[24,25], but the shielding effect of the surface crust would seriously hamper the reaction. Few experimental studies have explored how amino acids were aqueously altered in the presence of minerals and under varying redox conditions at low temperature (≤150 °C).

Here, we hypothesize that low-temperature geo-electrochemical reactions altered amino acids in water-rock differentiated CCs' parent

[1]Earth-Life Science Institute, Tokyo Institute of Technology, 2-12-1-IE-1 Ookayama, Meguro-ku, Tokyo, Japan. [2]Super-cutting-edge Grand and Advanced Research (SUGAR) Program, Institute for Extra-cutting-edge Science and Technology Avant-garde Research (X-star), Japan Agency for Marine-Earth Science and Technology (JAMSTEC), Yokosuka, Japan. [3]Institute of Nature and Environmental Technology, Kanazawa University, Kanazawa, Japan. [4]Department of Chemistry and Chemical Biology, Rensselaer Polytechnic Institute, Troy, NY, USA. ✉e-mail: yamei.li@elsi.jp

bodies (icy planetesimals) due to pH and redox gradients. In the early solar system, icy planetesimals may have harbored pH and redox gradients induced by water-rock interaction[26,27]. The progressive hydrothermal alteration of Fe (II) primitive minerals (such as olivine) in a porous rocky core should have generated $H_2$-rich fluids at moderately high temperatures owing to the decay of short-lived radionuclides[28,29]. The secondary mineral assemblages are rich in serpentine, saponite, and metal sulfides, which keep the fluid pH alkaline in the core (e.g., pH 9~13)[28,29]. On the other hand, if a CC's parent body formed beyond the snowline of $CO_2$ in the protoplanetary disk[30,31], an overlying subsurface ocean would have contained high levels of dissolved carbonate with circumneutral to weakly alkaline pH (∼7–10) and be oxidative compared with the $H_2$-rich fluids in the core[30]. The pH and redox gradients in icy planetesimals were expected to be similar to that in Enceladus based on their similar rock and ice compositions[32–34]. Even for undifferentiated parent bodies, pH and redox gradients can occur given the heterogeneity of water/rock interaction conditions[35] associated with the compositional variations of chondrules[36]. Notably, such gradients in pH and redox potentially drive organic conversions via "geo-electrochemical processes" (GEPs) (Fig. 1). Electrons generated by $H_2$ oxidation transfer to the exterior surface to alter amino acids (AAs). CCs contain metals and semiconducting sulfides[37,38] and show hopping-type impurity conduction or semiconduction[39], rendering the ability for long-range electron transfer. GEPs have been observed in deep-sea hydrothermal vents on Earth[40,41], artificial metal sulfide hydrothermal chimney systems[42], and can activate the reduction of $CO_2$ and $NO_3^-/NO_2^-$ using the reducing energy of $H_2$[43–46]. On Mars, GEPs were proposed for organic synthesis through $CO_2$ reduction on Fe-rich minerals[47].

Here we examine the alteration of amino acids under simulated electrochemical conditions within icy planetesimals. Three proteinogenic amino acids (glycine (Gly), alanine (Ala), and valine (Val)) that were commonly discovered in CCs[48–50] were tested. We demonstrate that the electrolysis of these amino acids simultaneously generates monoamines, α-hydroxy acids, and monocarboxylic acids under a wide range of redox conditions at ambient temperature (25 °C). This process can account for the depletion of amino acids in heavily altered CCs as well as the relative enrichments of monoamines and hydroxy acids there. Our results thus suggest that $H_2$ is an important driver for the amino acid alteration via electrochemistry in aqueously altered CCs.

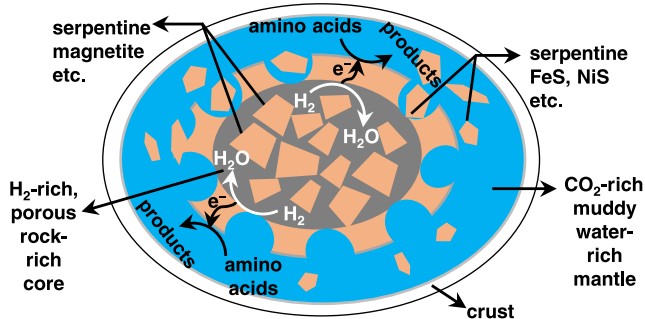

**Fig. 1 | Model of geo-electrochemical alteration of amino acids induced by water-rock interaction in icy planetesimals (a parent body of CCs).** A difference in the water/rock mass ratio (W/R) between the core (gray) and mantle (blue) leads to gradients in pH and redox. In a porous, rock-rich core with a low W/R ratio (<1)[55,56], the fluid is buffered to alkaline (pH 9-13) with serpentine and magnetite as the dominant minerals. The fluid contains abundant $H_2$, which serves as an electron source. Oxidation of $H_2$ to proton/water generates electrons which transfer through electrically-conductive minerals and trigger the catalytic, reductive alteration of amino acids, at the interface of sulfide mineral and the fluids in the muddy, water-rich mantle.

## Results

### Alteration of amino acids under simulated electrochemical conditions

Electrolysis of amino acids was conducted at ambient temperature (25 °C) in an electrochemical cell made of two compartments separated by a proton-exchange membrane. In the cathodic chamber, metal sulfide catalysts were placed on a carbon paper electrode in a pH-7 phosphate buffer containing 20 mM of amino acid. The concentration of amino acid is comparable to other studies where the stability of amino acids was investigated[51–53]. Except for achiral Gly, racemic mixtures of L- and D-enantiomers were used in the case of Ala and Val. The anodic chamber contains a platinum (Pt) electrode inserted in the pH-7 phosphate buffer alone. An anoxic condition was maintained by continuous argon gas flow during electrolysis (see the electrochemical setup in Supplementary Fig. 1). Iron and nickel sulfides are ubiquitously observed in CCs[37]. These metal sulfides mediate multielectron transfer and facilitate redox conversions[43,45,46,54]. In this study, FeS and NiS were synthesized by fresh precipitation of sodium sulfide and iron or nickel chlorides, respectively, under anoxic conditions. The synthesized catalysts were identified to be mackinawite (FeS) and NiS of low crystallinity with weak X-ray diffraction peaks (Supplementary Fig. 2). Using these two sulfides as catalysts, the conversion of amino acids to various products, including primary amines, α-hydroxy acids, and monocarboxylic acids was observed after 14-day electrolysis of amino acids at −0.5~−1.0 V versus standard hydrogen electrode (vs. SHE). Here, the half reaction of electrochemical alteration of amino acid (Fig. 1) was simulated using electrolysis under a constant potential. The electrode potential represented the magnitude of the redox gradient ($H_2$ enriched core versus bicarbonate-buffered mantle) and was calculated by Nernst equation:

$$E_h = \frac{1}{2F}\left(2RTln\alpha_{H^+} - RTln\alpha_{H_2} - \triangle_f G^0(H_2)\right) \qquad (1)$$

The electrode potential is equivalent to the reduction potential of $H_2/H^+$ redox couple under specific pH and temperature conditions as depicted in Fig. 2a. A more negative electrode potential corresponds to a condition with more alkaline pH or higher temperature. We simulated the electrochemical potential range generated at a pH range of 9~13 and temperature range of 0~150 °C in the icy planetesimal core (highlighted in blue in Fig. 2a), where $H_2$ oxidation was redox coupled with the reductive alteration of amino acids. Considering a water/rock mass ratio lower than one in a porous rocky core[55,56], a pH range of 9~13 was estimated based on the thermodynamic simulation with an initial fluid composition of cometary fluid[28]. The mantle fluid of planetesimals is likely pH-neutral due to the buffering of bicarbonate[30]. Here, a phosphate buffer (pH = 7) was used to avoid any carbonaceous compounds generated from bicarbonate or carbonate. Bicarbonate/carbonate does not change the result significantly, as will be shown later. Figure 2b–d show the integrated analytical results for the organic products generated after 14-day electrolysis of Gly (blue), Ala (red), and Val (orange) at –0.9 V. Primary amines, including methylamine (**2a**), ethylamine (**3a**), and isobutylamine (IBA, **4a**), were detected (Fig. 2b) from the electrolysis of Gly, Ala, and Val, respectively. Along with the amine generation, α-hydroxy acids including glycolate (**1b**), lactate (**2b**), and 2-hydroxy-3-methylbutyrate (HMA, **5b**) were also detected (Fig. 2c). Ammonia (**1a**), methylamine (**2a**), and monocarboxylic acids including formate (**3b**) and acetate (**4b**) were detected as common products (Fig. 2b, c). Electrolysis of Val also generated glycolate (**1b**) (Fig. 2c, orange). The generation of all these organic compounds was further confirmed by $^1H$-NMR (Fig. 2d). Neither amine nor α-hydroxy acid contaminants were detected in the starting materials (Supplementary Fig. 19) and control experiments without amino acids (Supplementary Fig. 20). The amounts of formate and acetate contaminants formed through the 14-day electrolysis were 9.7 ± 0.4

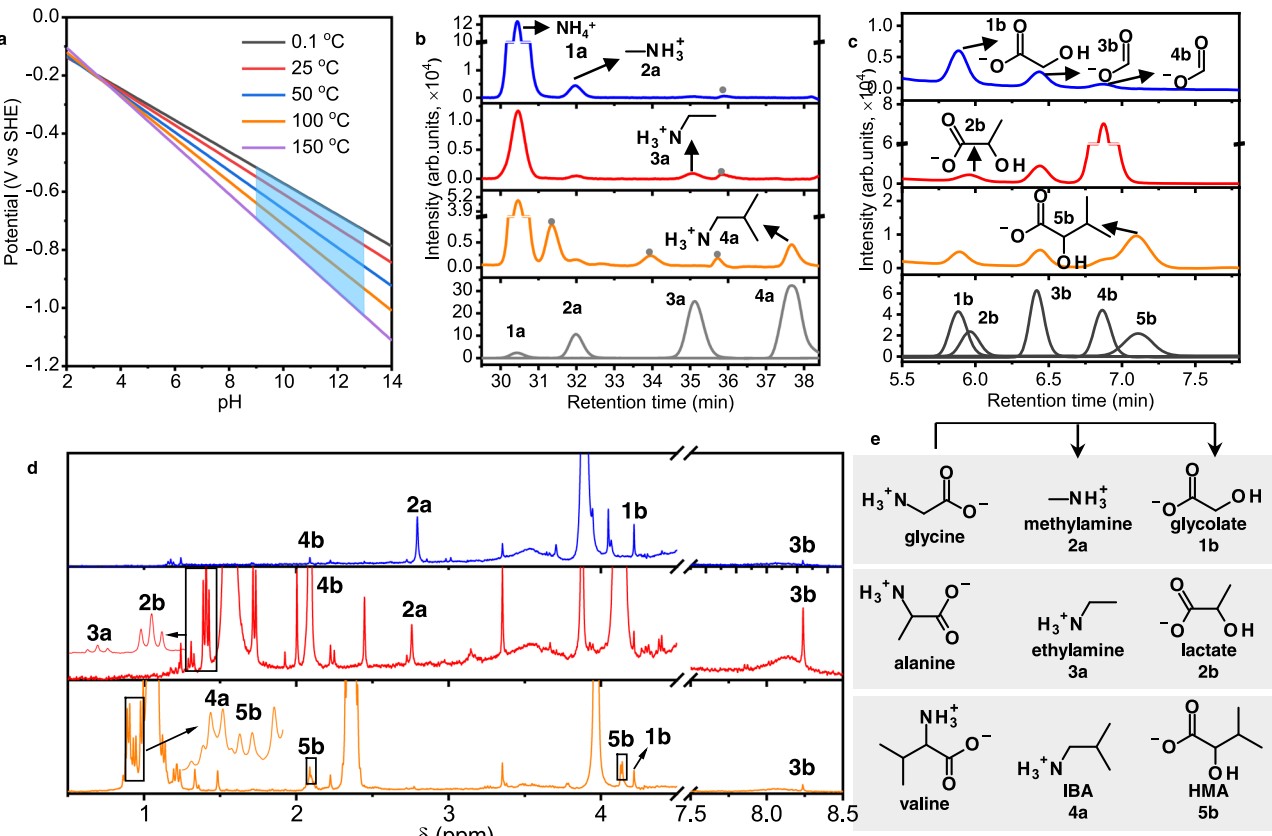

**Fig. 2 | Electrochemical potentials estimated in icy planetesimals and product detection after the electrolysis of glycine, alanine, and valine on FeS or NiS catalysts at ambient temperature (25 °C). a** The electrochemical potentials thermodynamically estimated at a wide pH and temperature range of water-rock interacted environment, where the potential range accessible in the core of icy planetesimal (parent body of CCs) was highlighted in blue. The potential was thermodynamically calculated by assuming $H_2$ oxidation as the half-reaction ($2H^+ + 2e^- \rightarrow H_2$) coupling with the amino acid alteration. A pH range of 9-13 in the core was assumed after the water-rock interaction based on the thermodynamic calculation by Neveu et al.[28]. **b** Chromatograms showing ammonia and primary amine products after electrolysis of glycine (blue), alanine (red), and valine (orange). The amine standards (bottom panel, gray) were designated by peak numbers (**1a–4a**). **c** Chromatograms showing the organic acid products after electrolysis of glycine (blue), alanine (red), and valine (orange). The carboxylic acid standards (bottom panel, gray) were designated by peak numbers (**1b–5b**). **d** $^1$H-NMR spectra of the product solutions after electrolysis of glycine (blue), alanine (red), and valine (orange). All the chromatograms were water-corrected. FeS (red and orange) and NiS (blue) were used as the catalyst at –0.9 V. Peaks denoted in dark gray dots in **b** were not assigned due to a lack of standards. The molecular structures of related compounds were shown in **e** with peak numbers and names. Abbreviations: IBA isobutylamine, HMA 2-hydroxy-3-methylbutyrate.

and 7.2 ± 0.5 μM, respectively. The reported concentrations were corrected by subtracting these blank values.

## Potential- and catalyst-dependent generation of amines and α-hydroxy acids

All the amine and α-hydroxy acid products have been detected in various CC subgroups[16,57–60]. Results on the potential- and catalyst-dependent generation of these compounds are presented in Fig. 3. Gly electrolysis generated both methylamine (Fig. 3a) and glycolate (Fig. 3b) under all tested conditions, suggesting the facile nature of these reaction pathways. Both of these products are preferentially generated on FeS compared to NiS. The formation of amines (Fig. 3a, c, e) was catalyzed by both FeS and NiS; while α-hydroxy acids tended to be more preferentially generated on FeS. For example, Val electrolysis generated HMA only on FeS (Fig. 3f). Regarding the potential dependence, methylamine and glycolate reach the maximal yields at the most negative potential (−1.0 V) (Fig. 3a, b). Generally, the electrochemical reaction rate increases exponentially upon the elevation of the thermodynamic driving force $-nF(E - E^0)$, where n is the number of electrons transferred, $F$ is the Faraday constant, and E and $E^0$ are the applied potential and standard reduction potential, respectively. Although the Gly-to-glycolate conversion does not formally involve

electron transfer, the observed potential dependence is possibly due to the generation of reduced Fe or Ni active species at more negative potentials[54,61]. Other amines and α-hydroxy acids sourced from Ala and Val did not show regular potential dependence likely due to the more complex network of competing reactions. A control experiment without a metal sulfide catalyst does not generate methylamine and glycolate from Gly (Supplementary Fig. 21), suggesting the key role of sulfide catalysts. Moreover, methylamine and glycolate were generated on both FeS and NiS under a diluted Gly concentration (0.2~2 mM) (Supplementary Fig. 14). This indicates that electrochemical alteration of Gly can proceed under a wide range of substrate concentrations. Notably, glycolate yield increases upon elevating the starting Gly concentration, suggesting that Gly molecule was involved in the rate-limiting step of glycolate production pathway. Unexpectedly, methylamine showed increased yield with lower starting Gly concentration. This result indicated that methylamine generation competed with glycolate generation and was less dependent on the Gly concentration. To simulate the bicarbonate-buffered muddy mantle, an additional experiment was conducted in a bicarbonate buffer (0.2 M, pH 8.2) at –0.9 V on NiS. No remarkable change in the yields of methylamine and glycolate in response to the buffer replacement was observed (Supplementary Fig. 15). Notably, formate was

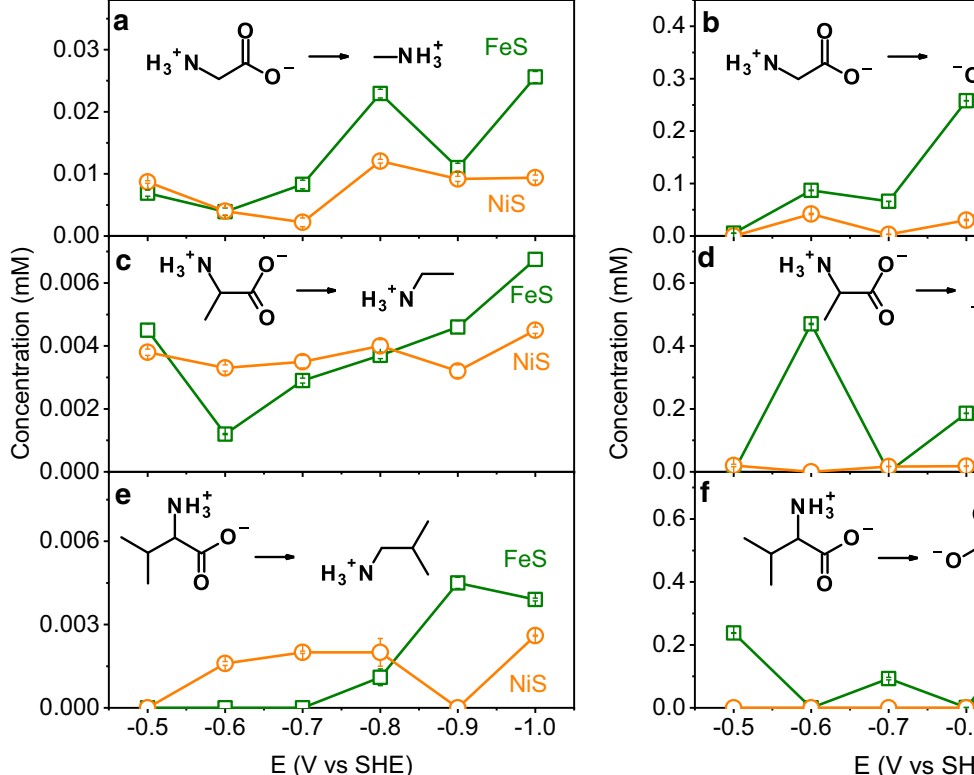

**Fig. 3 | Concentrations of amine and α-hydroxy acid products after two-week electrolysis of glycine, alanine, and valine at −0.5∼−1.0 V on FeS and NiS at 25 °C.** Molar concentrations of methylamine (**a**), glycolate (**b**), ethylamine (**c**), lactate (**d**), isobutylamine (IBA, **e**), and 2-hydroxy-3- methylbutyrate (HMA, **f**) generated from electrolysis of glycine (**a**, **b**), alanine (**c**, **d**), and valine (**e**, **f**) under different potentials (–0.5∼–1.0 V vs SHE) using FeS (green lines) or NiS (orange lines) as a catalyst. See Supplementary Figs. 6 and 7 for the HPLC chromatograms of product solutions and Supplementary Figs. 8–13 for their $^1$H-NMR spectra. The error bars are the standard deviations associated with the repeated measurements.

generated with a higher yield upon this buffer change (Supplementary Fig. 15), which is in accordance with the generation of formate via $CO_2$ reduction in a $CO_2$ saturated solution in our previous study[43].

**Reaction pathways**

Product concentrations under all examined conditions are summarized in Supplementary Tables 1–3. Compared to Gly, products were more complex in the case of Ala and Val, due to the generation of formate, acetate, methylamine, and/or glycolate as common products (Fig. 2b–d, Supplementary Tables 1–3). All the detected compounds are presumably generated via either decarboxylation or deamination (Fig. 4a). Amines (**2a**, **3a**, **4a**) were likely generated via decarboxylation, with formate as a possible byproduct (Fig. 4a, pathway I). α-hydroxy acids (**1b**, **2b**, **5b**) were presumably generated through deaminative C-N bond-breaking (Fig. 4a, pathway II), with ammonia as a byproduct. To gain more insight into the reaction pathways to methylamine and glycolate from Gly, Gly with the α-C labeled with $^{13}$C was used to trace the bond-breaking steps. Various $^{13}$C-enriched compounds with expected peak splitting patterns and $^{13}$C-$^1$H spin coupling constants were detected by $^1$H-NMR (Fig. 5a, bottom), and the spectrum was compared with that generated from Gly without isotope labeling (Fig. 5a, upper). Methylamines containing either $^{12}$C- or $^{13}$C-methyl groups were detected (Fig. 5a). $^{13}$C-methylamine generation confirmed the decarboxylation pathway, with a 2e$^-$ reduction to generate $^{12}$C-formate (Fig. 4b, pathway V). Additionally, $^{12}$C-methylamine generation indicates another pathway of $^{12}$C-carbonyl reduction to $^{12}$C-methyl with amination (Fig. 4b, pathway VI). The reduction could proceed via a formyl intermediate based on the detection of $^{12}$C-formate. $^{13}$C-glycolate was detected, confirming the deamination mechanism (Fig. 4b,

pathway IV). Besides $^{12}$C-formate, $^{13}$C-formate was generated based on the observed doublet (Fig. 5a, bottom). The oxidation of α-C was unlikely under reductive and anoxic experimental conditions. A possible pathway starts with the elimination of $^{13}$C-Gly to $^{13}$C-cyanide (CN$^-$) (Fig. 4b, pathway VII), which can subsequently hydrolyze into $^{13}$C-formate through formamide intermediate (pathway VIII) as reported recently[62]. Pathway VII has been reported in pyrolysis[63] and enzymatic[64] systems. No $^{13}$CN$^-$ was detected using $^{13}$C-NMR possibly due to a very low concentration arising from the rapid consumption of this active intermediate. The electrolysis of CN$^-$ at pH 7 at −0.9 V generates methylamine and formate (Supplementary Fig. 16), which thus suggests the feasibility of $^{13}$CN$^-$ as an intermediate for generating $^{13}$C-formate and $^{13}$C-methylamine.

Notably, regardless which of the amino acids were used as the starting material, formate, acetate, methylamine, and/or glycolate were generated (Fig. 2, Supplementary Tables 1–3), which suggests a Gly-like intermediate generated through Ala and Val electrolysis (Fig. 4a, pathway III). Indeed, Gly was detected as a product from Ala electrolysis (Fig. 5b). Gly-like intermediate likely played an important role during Ala electrolysis on NiS, as demonstrated by the generation of acetate as a major product (86∼92% of the total organics). In Murchison, the abundance of carboxylic acids is higher than amines, hydroxy acids, and amino acids by one order of magnitude[1], with acetate as the most dominant monoacid[65,66]. Our results thus suggest the importance of metal sulfides for catalytically generating meteoritic carboxylic acids. We note that ammonia was generated in a much higher yield than that of α-hydroxy acid probably because of the generation of other carbonaceous products unidentifiable due to the lack of standards or overlap with the signal of amino acids as shown in the $^1$H-NMR spectra in Supplementary Figs. 8–13. No chiral selection

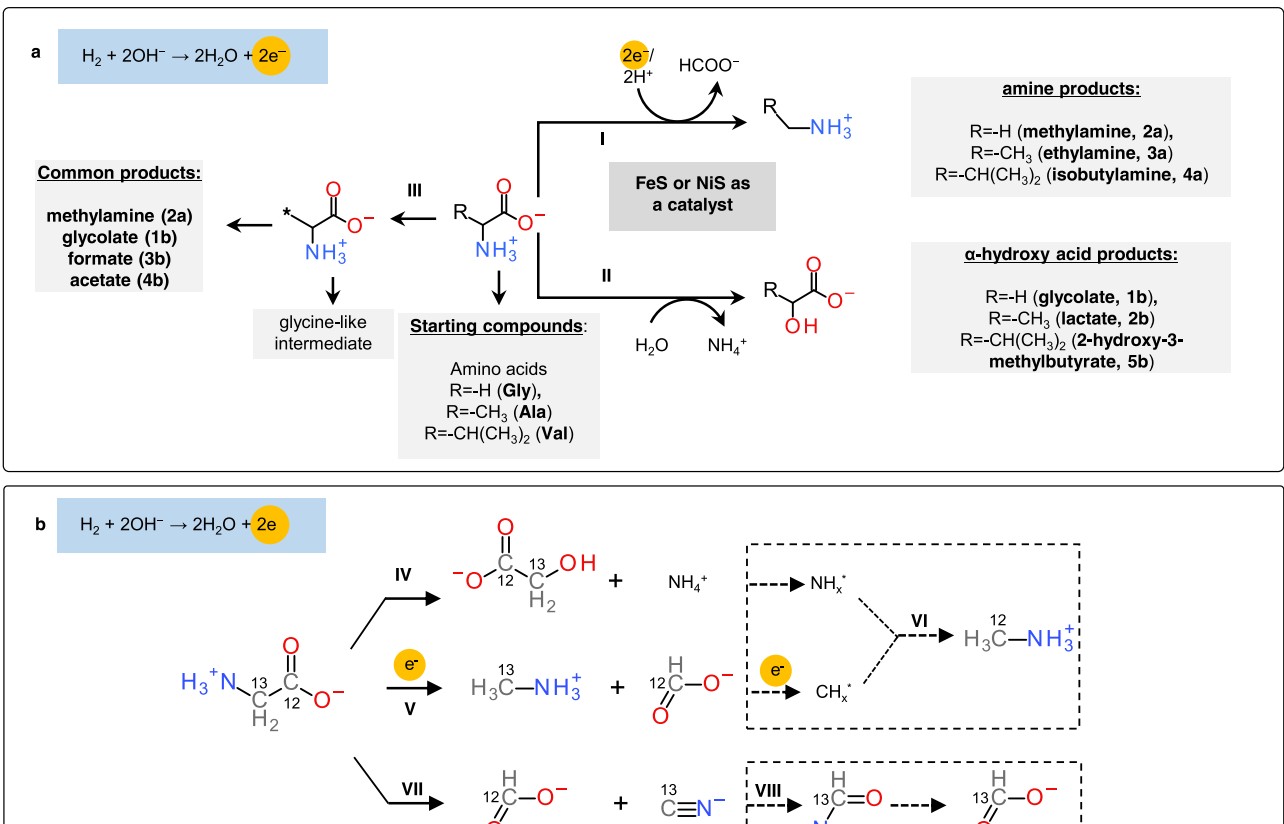

**Fig. 4 | Reaction pathways of electrochemical alteration of amino acids.**
**a** Shows the presumed pathways. Starting from amino acids, derivative products of amine and α-hydroxy acids were generated via decarboxylation (pathway I) and deamination (pathway II), respectively. Pathway III shows the reaction mediated by a glycine-like intermediate, leading to the generation of common products.

**b** Summarizes the pathways involved in glycine electrolysis deduced based on isotope labeling results. Steps denoted in dashed lines and arrows were based on hypothetical intermediate states and literature report[62]. $NH_x*$ and $CH_x*$ represent the adsorbed intermediates. The electrons participated in these reactions are sourced from $H_2$ oxidation reaction and highlighted in yellow.

was observed in our experiment as confirmed by HPLC using a chiral column (Supplementary Fig. 17).

To explore the universality of the reaction mechanisms of α-amino acids shown in Fig. 4a, aspartate (Asp) and glutamate (Glu) with anionic side chains were used as the starting substrates (20 mM) for electrolysis at −0.9 V on either FeS or NiS. The chromatograms of product solutions were shown in Supplementary Figs. 22–25. Starting with Glu, γ-aminobutyrate and α-hydroxyglutarate were generated, presumably through decarboxylation and deamination processes. Meanwhile, starting from Asp, β-alanine and malate were generated presumably via similar pathways. Additionally, glycine, methylamine, ammonia, acetate and formate were detected as common products. This indicates that electrolysis of Asp and Glu also contains reaction pathways mediated via Gly-like intermediates. Taken together, the reaction pathways illustrated in Fig. 4a can be extended to the α-amino acids with anionic side chains. The reaction pathways of electrolysis of Glu and Asp were summarized in Supplementary Fig. 26. Notably, Glu electrolysis also generated Asp and Asp-derived compounds (malate and β-alanine). This suggests another reaction pathway with a C-C bond breaking and recombination of a Gly-like and acetyl groups. The exact mechanism is unclear and will be investigated in future experiments.

## Discussion
Growing evidence has shown that amino acid abundances are greatly affected by aqueous alteration since 2007[12–14,67]. Among the CCs from the same subgroups (CM or CR), the most aqueously altered samples showed much lower amino acid abundances than more primitive ones

both at the total and individual levels[12–14] (Supplementary Table 4). This suggested that amino acids were destroyed by aqueous alteration[13,14]. Nevertheless, compared to the intensively studied synthetic routes of amino acids[5–7,68,69], alterations of amino acids haven't been experimentally demonstrated under low-temperature ranges of aqueously altered CCs (0~150 °C) to our knowledge. Our results demonstrated electrochemical alteration of amino acids proceeds through low-temperature (25 °C), catalytic processes on FeS and NiS surfaces. The chemical energy stored in reducing $H_2$ can serve as an alternative energy source in icy planetesimals, and $H_2$ in deep-sea hydrothermal systems can drive many C, N-related conversions via electrochemistry by coupling $H_2$ oxidation with the reduction of oxidative species (e.g., $CO_2$, $NO_3^-$, $NO_2^-$)[43,45,46,61]. In situ geo-electricity flow has been demonstrated in deep-sea hydrothermal vent systems where the hydrothermal vents were enriched with $H_2$ gas[40,41]. The threshold potential for activating amino acid alteration is −0.5 V, which corresponds to pH 8.7 at 0.1 °C, pH 7.5 at 50 °C, pH 7.1 at 100 °C or pH 6.7 at 150 °C at an $H_2$ concentration of 1 mmolkg$^{-1}$ at 100 bar (Fig. 2a). Alkaline pH and higher temperature can readily generate potentials more negative than −0.5 V. An elevated degree of aqueous alteration is associated with a higher water/rock ratio[55,56], and/or a prolonged duration of water-rock interaction, by which electrochemical alteration of amino acids would be further promoted. Geo-electrochemical decomposition of amino acids as a counterbalance to the aqueous synthesis of amino acids thus provides a possible solution to the "water paradox".

Methylamine, ethylamine, glycolate, and lactate are present as the dominant amine and hydroxy acid species in CCs[16,57,58,70]. In this study, electrochemically generated amine products show decreased

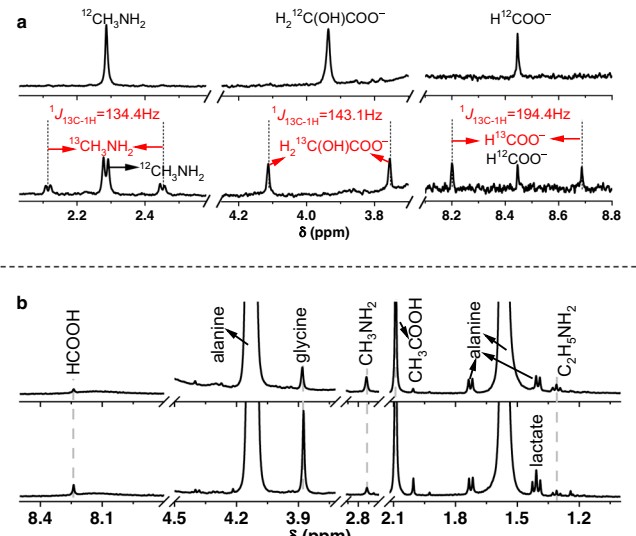

**Fig. 5 | ¹H-NMR spectral evidence of reaction pathways described in Fig. 4.**
**a** Shows the ¹H-NMR spectra for the detection of methylamine ($CH_3NH_2$), glycolate ($H_2C(OH)COO^-$), and formate ($HCOO^-$) from electrolysis of glycine with α-C labeled by ¹³C (bottom) and without isotope labeling (upper) on FeS at −1.0 V. **b** Shows the ¹H-NMR evidence of glycine formation by alanine electrolysis at −0.9 V (bottom) and −0.8 V (upper) on NiS catalyst.

concentrations with increasing carbon numbers in the homolog series ([methylamine] > [ethylamine] > [isobutylamine]) under all tested potentials on both FeS and NiS (Fig. 6a). The homolog tendency was conserved at lower amino acid concentrations (2 mM) (Supplementary Fig. 18). Such homolog tendency is consistent with that in aqueously altered CI, CM, and CR chondrites (Fig. 6b)[13,16,57]. From a mechanistic point of view, the decarboxylation rate is expected to decrease with a longer side chain length due to the increased polarity of the α-C-carbonyl carbon bond through the inductive effect[20]. Additionally, methylamine generation via a Gly-like intermediate from all three amino acids further increases the relative abundance of methylamine (Figs. 2b, 4a, pathway III). The synthetic origins of CCs' amines are controversial. Methylamine and ethylamine have been synthesized in a simulated cosmochemical process from a mixture of ammonia and methane model ice[71]. Recently, a Formose-type aqueous reaction of formaldehyde, glycolaldehyde, and ammonia also generated methylamine and ethylamine accompanied by Gly and Ala[7]. Methylamine and ethylamine were indeed observed in comets[71]. Nevertheless, monoamines with a backbone carbon number higher than two have not been experimentally demonstrated previously, despite their pervasive presence in CCs[16,57,58,70]. Here, isobutylamine with a backbone carbon number of four was generated via electrolysis of Val, indicating that electrochemical alteration of amino acids can serve as a potentially universal pathway to generate various amines with longer carbon chains from their amino acid analogs. The effective generation of amines by aqueous, electrochemical processes can also explain the 1–3 orders of magnitude higher amine abundances in aqueously altered CI, CM, and CR groups than those in pristine CO, CV and thermally metamorphized CK ones with little or no aqueous alteration[70].

α-hydroxy acids were building blocks of the polyester droplets, serving as a type of proto-compartments for encapsulating and stabilizing biomolecules[72,73] and promoting prebiotic reactions[74] in protocells. α-hydroxy acids in CCs show a molecular distribution comparable to those of amino acids and therefore were suggested to be aqueously synthesized from common aldehyde/ketone precursors with amino acids via hydrogen cyanide and ammonia addition (Strecker-cyanohydrin reaction)[59]. Alternatively, α-hydroxy acids can be synthesized by reduction of α-keto acids by

Fe hydroxides[8,9]. Nevertheless, the discovery of only two α-keto acids (pyruvate and α-keto glutaric acid) in meteorites[75] suggests that reduction of α-keto acid cannot provide a reliable pathway to generate all the α-hydroxy acids with comparable isomer and analog varieties in CCs. Glycolate and lactate are the most abundant hydroxy acids in CR and CM chondrites[13,60]. Here, electrochemical alteration of amino acids provided a potentially universal pathway to generate hydroxy acids from their amino acid analogs. In contrast to amines, electrochemically generated α-hydroxy acids didn't show consistent homolog tendencies (Fig. 6c), and the concentration of lactate was higher than glycolate at −0.6 V and −1.0 V on FeS. This follows a similar trend in Murchison (Fig. 6d). Meanwhile, a monotonic homolog tendency ([glycolate]>[lactate]>[HMA]) observed at other potential conditions is similar to two CR2 chondrites (LAP 02342 and GRA 95229)[60] (Fig. 6d). A higher concentration of glycolate than lactate was also observed in various CR chondrites, with two exceptions (QUE 99177 and MIL090001). The non-monotonic homolog tendency in the α-hydroxy acids series is possibly due to the complex reaction kinetics affected by multiple competing reaction pathways (Fig. 4a).

Amino acids were either synthesized or decomposed during the aqueous phase in CCs' parent body; however, it is still challenging to determine which process is dominant based on organic records. Based on the electrochemical reaction networks associating amino acids with their amine and α-hydroxy acid derivatives, the effect of aqueous alteration on their abundances in CCs should be evaluated by integrated analyses, rather than focusing on the amino acid concentration alone[13,14]. Figure 7a–c depicted the abundances of Gly, Ala, and Val and their amine and α-hydroxy acid analogs in six CR chondrites with varying petrology as reported recently[13]. All amino acids show the lowest abundances while hydroxy acids show the highest abundances in the most heavily altered GRO 95577/CR2.0 and MIL 090001/CR2.4 samples. Such trade-off behavior between amino acids and hydroxy acids, again, suggests the conversion from amino acids to hydroxy acids. Apparent differences in amino acid abundances among two CR2.7 and two CR2.8 chondrites were shown, likely due to a heterogeneous accretion[13]. This suggests the absolute concentration of amino acids is not an optimal descriptor of the aqueous alteration degree. Figure 7d–f plotted the molar ratios of amine/amino acid and hydroxy acid/amino acid. Notably, the data showed a clear divergence between heavily altered and primitive chondrites, as highlighted in blue and red, respectively. The primitive chondrites show very low ratios ranging from 0.02 to 2.17 (tabulated in Supplementary Table 7). The ratio ranges of hydroxy acid/amino acid are close to the Strecker-cyanohydrin synthesis (processes 1, 2 in Fig. 7g), with reported ratios of [Glycolate]/[Gly] and [Lactate]/[Ala] of 0.73 and 0.33, respectively[6]. The ratio ranges of amine/amino acid are close to the Formose-type synthesis (processes 3 and 4 in Fig. 7g)[7], which generates molar ratios of methylamine/Gly and ethylamine/Ala of 0~0.33 and 0~0.1 at 20~120 °C, respectively. Therefore, aqueous synthesis through these two mechanisms would reasonably account for the relatively constant and low ratios in these four primitive chondrites. In comparison, in the heavily altered CR2.0 and CR2.4 samples, amine/amino acid and hydroxy acid/amino acid ratios among both Gly (Fig. 7d) and Ala (Fig. 7e) related compounds show markedly higher values than the primitive ones by 2~4 orders of magnitude. The markedly higher ratios are consistent with the accumulation of amines and hydroxy acids at the expense of amino acid analogs with elongated alteration. This result thus strongly suggests the dominant role of amino acid alteration (processes 5 and 6 in Fig. 7g) in shaping the distributions of these associated compounds in heavily altered samples. Moreover, hydroxy acids show higher abundances than amines (Fig. 7a, b) in these heavily altered samples, which is in agreement with the higher yield of hydroxy acids than amines in electrochemical systems (Supplementary Tables 1 and 2).

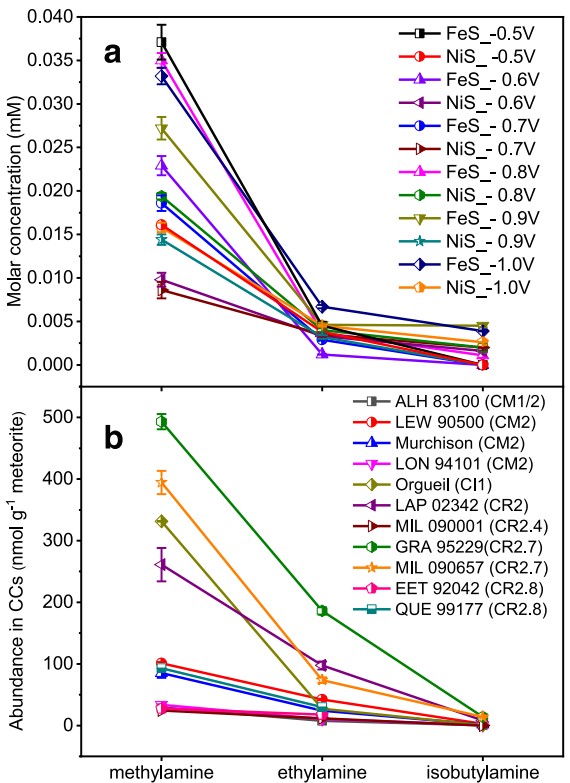

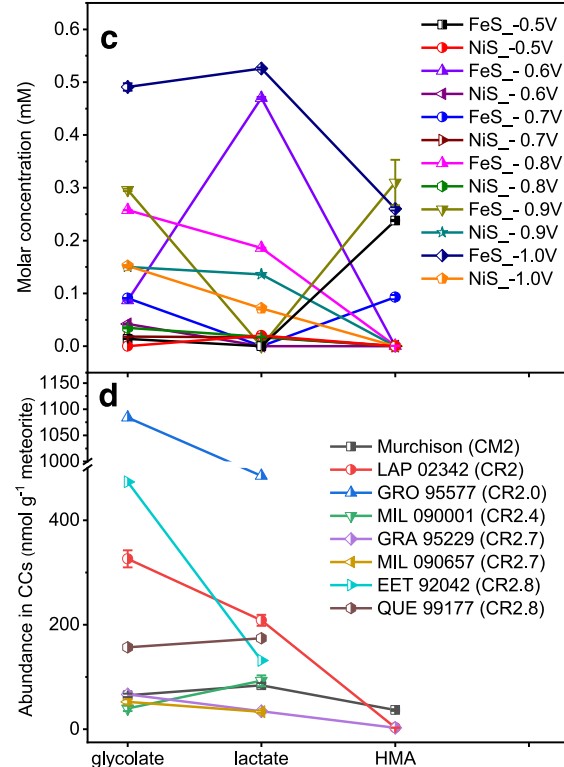

**Fig. 6 | Homolog tendencies of amines and α-hydroxy acids generated via electrochemical alteration of amino acids and their comparison with CCs.**
**a** The molar concentrations of methylamine, ethylamine, and isobutylamine formed by electrolysis of glycine, alanine, and valine, respectively, under various potential conditions on FeS or NiS. Since all these three amino acids generate methylamine, the presented methylamine concentration is the sum of values from all the three amino acid electrolysis experiments operated under the same condition. The error bars are the standard deviations associated with the repeated measurements. **b** The amine concentrations in CM, CI, and CR

chondrites (Murchison[57], three other CM meteorites (ALH 83100, LEW 90500, and LON 94101)[16], Orgueil[57], and six CR meteorites[13, 16]). **c** The molar concentrations of glycolate, lactate, and 2-hydroxy-3-methylbutyrate (HMA) generated by electrolysis of glycine, alanine, and valine, respectively. Since both glycine and valine generate glycolate upon electrolysis, the presented glycolate concentration is the sum of values from glycine and valine electrolysis experiments operated under the same condition. **d** The concentrations of these three α-hydroxy acids in Murchison, and CR meteorites[13, 60]. Data that were not reported were not shown in the plots.

Based on the clear divergence in the abundance ratios of methylamine/Gly, glycolate/Gly, ethylamine/Ala, and lactate/Ala between heavily altered and primitive CR chondrites (Fig. 7d, e), these ratios can serve as more predictive descriptors of aqueous alteration degrees and formation mechanisms, than the absolute abundances of any of these compounds[11]. The available data of HMA abundances[13] in Fig. 7c, f are too limited to make useful discussion. In Murchison (CM2.5), the abundance ratios of methylamine/Gly (2.12), glycolate/Gly (1.62), ethylamine/Ala (4.63), lactate/Ala (16.19), IBA/Val (0.47), and HMA/Val (11.01) also show values relatively higher than the primitive CR chondrites, indicating an alteration-dominated feature in its parent body. However, these data were calculated based on separate measurements on three different samples[14,57,60], therefore, mistakes could occur due to variations in the samples[59,60]. The amine and hydroxy acid abundances haven't been reported in other more primitive CM samples, such as the most primitive Asuka 12236 (CM2.9) sample[14]. Therefore, integrated measurements of amino acids, amines, and hydroxy acids in the same samples with a wide range of aqueous alteration degrees would be necessary for elucidating the origins and synthetic relations of these organics shaped by aqueous alteration in CM chondrites. Moreover, CCs without high-temperature heating will be required for this purpose, as some CM[14] and CI[76] chondrites were reported to experience heating at 500~600 °C, which almost removed amino acids completely.

Based on the above discussion, the alteration of amino acids into amines and hydroxy acids likely played a dominant role in shaping the distributions of these compound classes in heavily altered CCs. Under

this scenario, amines and hydroxy acids were simultaneously produced from their common amino acid precursors. Therefore, the relative concentration of amine and hydroxy acid can be utilized as a kinetic probe of the energetic conditions of associated reaction networks. Figure 8 plotted the concentration ratios of glycolate/methylamine (GA/MA, Fig. 8a, b), lactate/ethylamine (LA/EA, Fig. 8d, e), and 2-hydroxy-3-methylbutyrate/isobutylamine (HMA/IBA, Fig. 8g, h) as a function of reduction potentials on FeS or NiS catalyst. All the GA/MA, LA/EA, HMA/IBA ratios were highly dependent on the reduction potential and the identity of the catalyst, suggesting the two divergent pathways to amine and α-hydroxy acid are mutually competing and kinetically regulated by reduction potentials. The reduction potential is primarily determined by the water-rock interaction parameters, including the compositions of starting materials, water/rock ratio, temperature, and pH (Fig. 2a). Such relation thus provides a basis to constraining the water/rock interaction conditions of CCs' parent bodies using hydroxy acid/amine ratio as a descriptor. As a proof of concept, these three ratios in the five CR chondrites with the petrology ranging from 2.0 to 2.8 were calculated based on the reported concentrations[13] and shown in Fig. 8c, f, i (the abundances and ratios are tabulated in Supplementary Tables 4–6 and 8, respectively). Except for the GA/MA ratio at −0.5 V during Gly reduction, all the ratios resulting from electrochemical alteration (ECA) are higher than 1 (Fig. 8a–g), which reflects a higher generation rate of hydroxy acid than that of amine. This is in accordance with all the ratios in two heavily altered CRs (GRO 95577 and MIL 090001), which are also higher than 1 and at similar levels of magnitude with ECA values. GRO

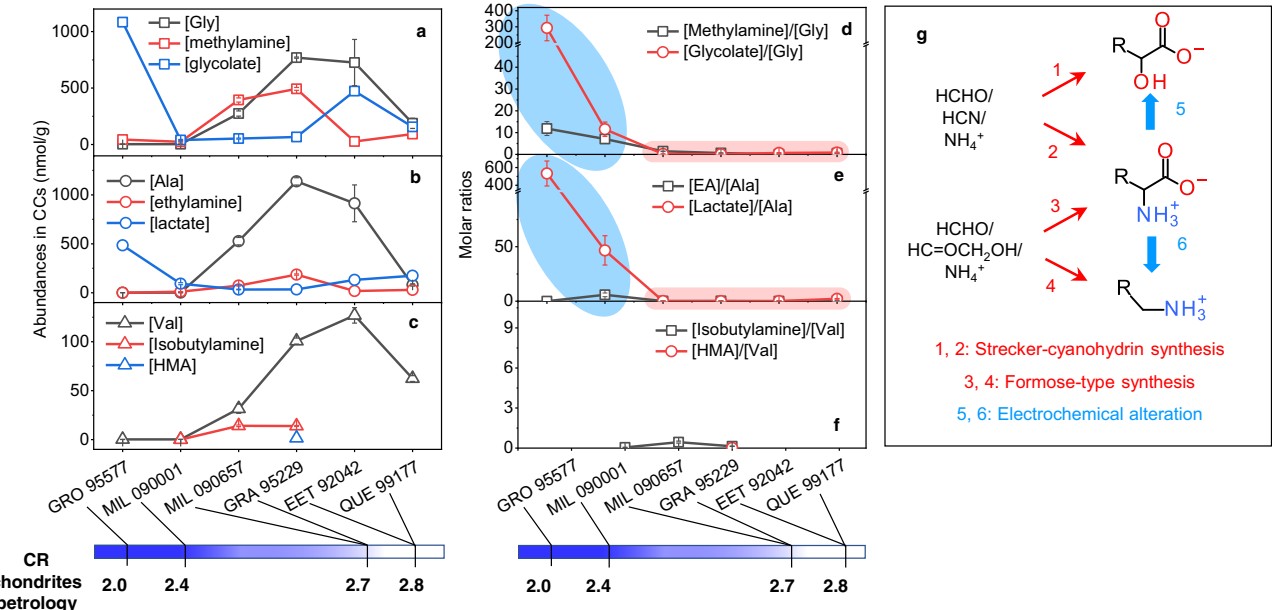

**Fig. 7 | Alteration-dominated versus synthesis-dominated features based on the organic distributions of amino acids, amines, and hydroxy acids in CR chondrites.** The molar abundances of glycine, alanine, and valine, and their structurally related amines and hydroxy acids were plotted in **a**, **b**, **c**, respectively. The abundances and the CR chondrites petrology depicted at the bottom were sourced from a recent report by Aponte et al.[13]. The abundances were tabulated in Supplementary Tables 4–6. An increased aqueous alteration degree was indicated by the increasing darkness of the blue color at the bottom bar. **d**, **e**, **f** plotted the molar ratios of amine/amino acid and hydroxy acid/amino acid among glycine-, alanine- and valine-related compounds, respectively. **g** schematically depicted the reactions for synthesizing amino acids (processes 2 and 3), amines (process 4), and hydroxy acids (process 1) in low-temperature, aqueous media (red arrows) and the electrochemical processes for altering amino acids into hydroxy acids and amines (blue arrows, processes 5 and 6). Abbreviation: HMA: 2-hydroxy-3-methylbutyrate.

95577/CR2.0 (denoted as GRO/CR2.0 thereafter) has the highest [GA]/[MA] ratio (24.75), which encompasses all the ECA values, while MIL 090001/CR2.4 (denoted as MIL/CR2.4 thereafter) shows relatively lower values (1.64). More selective catalysts favoring the deamination while suppressing decarboxylation would be necessary for generating the very high ratio in GRO/CR2.0, and vice versa. Alternatively, additional methylamine input from the decomposition of other amino acids (Fig. 4a, pathway III), cosmochemical or a Formose-type synthesis, will attenuate the ECA-ratio to the level in MIL/CR2.4. In regard to the lactate/ethylamine (LA/EA) ratio, no ethylamine was detected in GRO /CR2.0. The ratio in MIL 090001/CR2.4 is close to that obtained at −0.7 V and −0.8 V on NiS, and much lower than the values obtained on FeS. Supposing NiS is the dominant catalyst, this suggests that the T and pH of parent bodies of MIL/CR2.4 would be close to pHs of 11.5~13.3 at 25 °C, 9.9~11.2 at 100 °C, and 9.1~10.3 at 150 °C, to generate the potential of −0.7~−0.8 V (data can be extracted from Fig. 2a). Alternatively, supposing FeS is the dominant catalyst, to reproduce the relatively lower [LE]/[EA] ratio in MIL/CR2.4, combining different potential conditions is required. For example, Ala electrolysis at different potentials of −0.8 V and −0.6 V may attenuate the high ratio at −0.8 V to the level of MIL/CR2.4. Such different potential conditions are feasible since the aqueous alteration conditions have been suggested to vary from place to place in the parent body owing to heterogeneity in chemical compositions[35]. Aqueous alteration conditions also changed progressively with decreasing temperature due to the progressive decay of the radionuclides[77,78]. Considering a cosmochemical contribution of ethylamine[71], the electrochemically generated LA/EA ratio can further be attenuated to MIL/CR2.4 level. Notably, for the two more primitive CR2.7 chondrites (GRA 95229 and MIL 090657), both the MA/GA and EA/LA ratios are much lower than 1 (Supplementary Table 8 for the values). This suggests little effect of geo-electrochemical alteration on these less altered samples. Much higher ratio values can be seen in the most primitive CR2.8 samples (EET 92042 and QUE 99177), albeit little effect of geo-electrochemistry can be anticipated due to the limited aqueous activity. Finally, for the

[HMA]/[IBA] ratio, since HMA abundance was not reported in these samples, detailed discussion is impossible.

Besides the amino acid abundances, the distribution of amino acids tends to increase in favor of the non-α-amino acids upon an increased aqueous activity[12,13]. The increasing abundance of β-alanine relative to glycine in more aqueously altered meteorites could be related to a greater resistance of β-alanine to decarboxylation compared to glycine as demonstrated in high temperature (310~330 °C) hydrothermal experiments[20], which could also be the case in the electrochemical alteration system. Additionally, thermally altered meteorites, including CO3, CV3, Antarctic ureilites, R, and CK chondrites, have been reported to be dominated by straight chain, amino terminal (n-ω-amino) acids[79,80]. The range of temperatures that these meteorites have experienced is estimated to be 250~600 °C in CO and CV chondrites and 300~600 °C or warmer in R and CK ones. Although Fischer−Tropsch-type (FFT) synthesis has been argued to account for these phenomena, this argument faces challenges that 1) the very low amino acid abundances in specific CC types (CK and R chondrites) cannot be well explained; 2) experimental FFT syntheses have not been able to reproduce the similar organic distribution. Based on our model and experimental results, in addition to the synthetic pathway, the decomposition process should be carefully considered and included to account for the meteoritic amino acid distribution. The higher thermal stability of n-ω-amino acids than other isomers has been reported[52], due to the capability of these n-ω-amino acids to form intramolecular lactam condensates via dehydration. In addition, repositioning the NH₂ group from the α-carbon to the terminal carbon reduces inductive and electrostatic effects favoring decarboxylation[20]. Thus, the elevated relative abundances of n-ω-amino acids are possibly not mainly due to their preferential syntheses but that other isomers are more rapidly destroyed due to the lack of the ability to form stable lactam like products. In this regard, we believe that the detection of targeted decomposition products can shed further insight into this question[51].

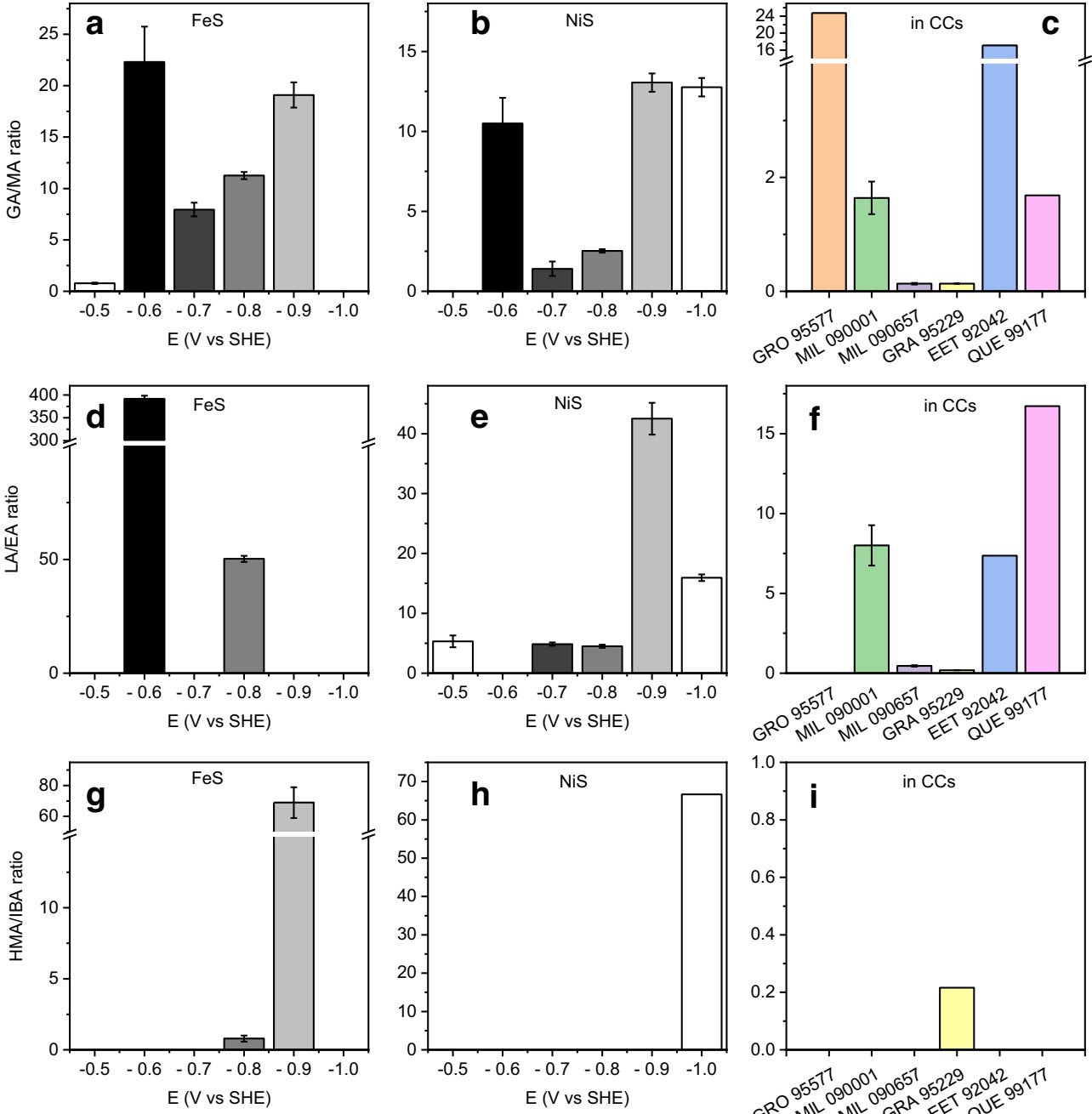

**Fig. 8 | Comparison of the molar ratios of glycolate/methylamine (GA/MA), lactate/ethylamine (LA/EA), and 2-hydroxy-3-methylbutyrate/isobutylamine (HMA/IBA) generated via electrochemical alteration of glycine, alanine, and valine, respectively, with the corresponding ratios in aqueously altered CR chondrites.** Molar ratios of GA/MA (**a, b**), LA/EA (**d, e**), and HMA/IBA (**g, h**) as a function of potential (−0.5–−1.0 V vs SHE) and type of catalyst (FeS: **a, d, g**; NiS: **b, e, h**), generated by electrolysis of Gly, Ala, and Val, respectively. For the values

not shown, the hydroxy acid was not detected. The corresponding ratios in six different CR chondrites (GRO 95577 (CR2.0), MIL 090001 (CR2.4), MIL090657 (CR2.7), GRA 95229 (CR2.7), EET 92042 (CR2.8), and QUE 99177 (CR2.8)) were calculated based on the reported concentrations[13] and depicted in **c** (GA/MA ratio), **f** (LA/EA ratio), and **i** (HMA/IBA ratio). Data not shown in **f** and **i** are due to the unreported concentration of ethylamine and HMA[13]. The error bars are the standard deviations associated with the repeated measurements.

Isovaline in primitive CR and CM chondrites show no measurable L-excesses within error[12,81]; however, the L-isovaline enantiomeric excess has been confirmed in various aqueously altered CM and CR chondrites, and its magnitude (ranging of 0.2~18.5%) increases with an increased extent of aqueous alteration[12,13,82]. This large enantiomeric excess cannot be easily explained by the photolytic decomposition by UV circularly polarized light in the presolar cloud, and its strong reliance on aqueous alteration degree indicates that L-isovaline enrichments occurred inside the parent bodies during aqueous alteration. Considering the universal conversion pathways

demonstrated in the electrochemical alteration system (Figs. 3 and 4), it is reasonable to expect that isovaline could be altered under similar electrochemical conditions. At the current stage, no chiral selection was observed for Ala and Val in the present electrochemical systems. The introduction of chiral catalysts or auxiliaries[83] may help with the creation and amplification of asymmetry with extended aqueous reactions under electrochemical conditions. Alternatively, electrochemically processed racemic amino acids can obtain chiral selection via other physical[84,85] or autocatalytic chemical[83] mechanisms.

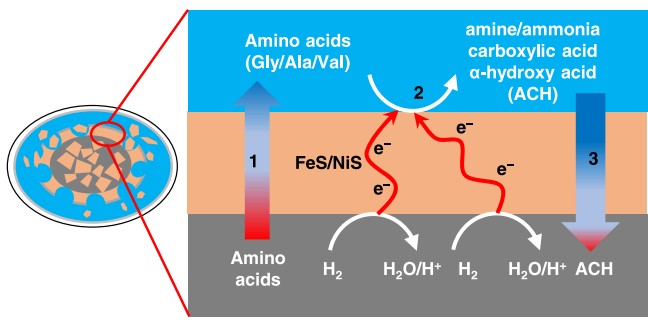

**Fig. 9 | Chemical recycling and circulation of soluble organics within icy planetesimals and other icy bodies.** 1: Through upwelling of hydrothermal fluids within the rock-rich core[88,89], the thermochemically produced amino acids[5] could have been transported to the interface with the overlying subsurface ocean; 2: Driving by the pH and redox gradient between the core and mantle fluids, amino acids can be electrochemically converted into amine, ammonia, carboxylic acids and α-hydroxy acids (ACH) driven by the reducing energy of $H_2$. 3: Part of these generated ACH compounds can be transported to the rock-rich core through fluid circulations within icy planetesimals[88,89].

We suggest the possibility of reaction cycles within icy planetesimals and other similar icy bodies (Fig. 9), such as Enceladus and early Ceres. Within a high-temperature rocky core, amino acids may have been generated through thermochemical aqueous reactions of formaldehyde, glycoaldehyde, and ammonia[5,86,87]. Through upwelling of hydrothermal fluids within the rocky core[88,89], the produced amino acids could have been transported to the interface with the overlying subsurface ocean (Fig. 9, process 1), where pH and redox gradients could have promoted geo-electrochemical processes. In the interface, the amino acids would have been, then, converted into derivative compounds, such as amine (including ammonia), α-hydroxy acids, and monocarboxylic acids (Fig. 9, process 2). Some of these soluble organics could have been, again, transported to the rocky core through fluid circulations within icy planetesimals[88,89] (Fig. 9, process 3). In these recycling systems, a variety of soluble organic matter can be generated due to differences in degrees of water circulation, alteration reactions, and temperatures of the rock-rich core.

We propose that the geoelectrochemical alteration of amino acids might be the origin of derivative organics and relevant molecules found on the solar system icy bodies, which can be tested with future in situ explorations. On Ceres, which is analogous to volatile-rich CCs[90], ammoniated saponite is known to be present widespread on the surface[91]. The surface of Ceres may reflect materials from an ancient subsurface ocean[92]. The Cassini spacecraft found the presence of complex organic molecules and ammonia in plume materials of Enceladus[34,93–95]. These bodies possibly accreted amino acids when they formed. Comets record the building blocks of the outer solar system bodies and are known to contain amino acids, which are thought to be interstellar in origin[96]. Spontaneous generation of amino acids in the interstellar medium is experimentally demonstrated to be possible by irradiation of interstellar ice containing primordial molecules by ultraviolet light[68,69] or high-energy protons[97]. This suggests the possibly wide distribution of amino acids in the outer solar system bodies due to the accretion of pristine ices. Amino acids were typically considered stable under low-temperature conditions (0–150 °C); however, our results suggest that redox gradients generated by water/rock interaction could degrade amino acids to lower molecular weight amines and organic acids (e.g., ammonia, methylamine, ethylamine, formate, acetate) (Supplementary Tables 1–3). McSween et al.[90] have proposed that Ceres' ammonia could have been derived from the decomposition of organics at high temperatures (~300 °C). Our study provides experimental support for this hypothesis, even at lower temperatures. Thus, we suggest that geo-electrochemical alteration of

organic molecules (e.g., amino acids) could serve as a source of ammonia in the interlayers of saponites on Ceres and in the plume of Enceladus. The Enceladus plume contains low-mass amines and carbonyls[93], which could have been partially derived from the degradation of amino acids via electrolysis as shown in this work. Provided that there exist redox gradients at their rocky core–water mantle boundaries[33,77], geo-electrochemical alteration of amino acids, if any, could possibly contribute to the detected organics and ammonia. We propose that further exploration missions for Ceres[98], Enceladus[99], and other icy bodies can test our geo-electrochemistry model. Sampling and analyzing their surface materials would confirm the existence of amino acids, derivative compounds (e.g., amines, hydroxy acids), and mineral catalysts. Supposing a relative enrichment of derivative compounds with respect to their amino acid analogs is detected in a sulfide-rich sample, this may indicate electrolysis of amino acids has operated and affected the organic distribution on these icy bodies.

## Methods
### Preparation of iron and nickel sulfides
Iron and nickel sulfides were prepared by drop-wise addition of 100 mM $Na_2S$ (aq) into a 100 mM aqueous solution of the corresponding metal chloride ($FeCl_2$, $NiCl_2$) under vigorous stirring with a final volume ratio of 1:1. To prevent oxidation by atmospheric $O_2$, the sample preparation was conducted in a glove box filled with Ar gas, with 3.97% $H_2$ being added (the COY system, COY Laboratory Products, United States). Solid precipitates were then separated from the supernatant by centrifugation and dried under a vacuum. The solid precipitates were ground into powders in a ceramic mortar before use. The synthesized sulfides showed broad x-ray diffraction signals, with low intensity indicating low crystallinity (Supplementary Fig. 2). All chemicals except for amino acids were purchased from Wako with reagent grade. Amino acids with L- and D-chirality were purchased from Peptide Institute. Inc., Japan. The purities of D-enantiomer in D-alanine and D-valine are higher than 99.97% and 99.99%, respectively. Deaerated Milli-Q water (18.2 megohms) was used as the solvent. 2-$^{13}$C-labeled glycine ($H_2NH_2{}^{13}C$-COOH) was purchased from Cambridge Isotope Laboratories, Inc. with a purity of >98%.

### Amino acid electrolysis experiments
Supplementary Fig. 1 shows a photograph of the electrochemical cell. For each experiment, about 200 mg of either FeS or NiS powders were deposited on the carbon paper working electrode (5.7 cm$^2$). Deaerated 100 mM phosphate buffer solution (mixture of $NaH_2PO_4$ and $Na_2HPO_4$, pH 7) was implemented into the cell (50 and 10 mL to the working and counter electrode chambers, respectively). One type of amino acid (Gly, Ala, or Val) was implemented into the cathodic chamber with a final concentration of 20 mM unless otherwise noted. Argon (Ar) gas was purged into the solution (20 ml min$^{-1}$) for at least 1 h prior to the electrolysis. After purging, the solutions were sampled as blanks. While keeping the Ar purging, a constant potential was applied on the working electrode by using a multi-channel potentiostat (PS-08; Toho Technical Research). All potentials were measured against an Ag/AgCl reference electrode in saturated KCl and converted to the SHE (standard hydrogen electrode) scale using the following equation[100]

$$E \text{ (vs. SHE)} = E \text{ (vs. Ag/AgCl)} + 0.197 \text{ V} \qquad (2)$$

After two weeks of duration, the sample solution was filtered with a polytetrafluoroethylene (PTFE) membrane filter (pore size, 0.22 um) and was analyzed by multiple analytical methods as described below. A control experiment without using a metal sulfide catalyst was conducted by two-week electrolysis of Gly (20 mM) in a phosphate buffered electrolyte (pH 7) at −0.8 V using a bare carbon paper electrode. Another control experiment was conducted by two-week electrolysis

of a phosphate buffered electrolyte (pH 7) free of amino acids at −0.9 V using FeS as a catalyst.

## Product analyses

Organic acid products were detected and quantified by using a Shimadzu HPLC system equipped with an electric conductivity detector and an anion exchange column (Shim-pack SCR-102H, Shimadzu) set at 40 °C. The p-toluenesulfonic acid solution (5 mM) was used as the eluent at a rate of 1.6 ml min⁻¹.

Primary amine, amino acid, and peptide products were analyzed using a Jasco HPLC system equipped with post-column derivatization with o-phthalaldehyde and a fluorescence detector operated at 345 nm for excitation and 455 nm for emission. Five citrate buffer solutions of different citrate concentrations and pH values were used as eluents in a stepwise manner. A cation-exchange column (AApak Na II-S2, Jasco) was used at 50 °C. No peptides from glycine were detected by HPLC in this study. Chiral analysis was conducted using an HPLC system installed with a chiral column (CROWNPAK® CR-I (+)) at 25 °C with a UV detector (detection wavelength is 200 nm).

$^1$H-NMR spectra were acquired using a Bruker Advance III spectrometer (400.1318 MHz) at the sample temperature of 303.0 K. Typically, 1 ml of sample solutions was tuned to pH ∼1 by adding 0.05 ml of 6 M HCl. 0.45 ml of pH-tuned sample solutions was mixed with 0.1 ml of $D_2O$ (99.9%; Merck Millipore) containing 5 mM 3-(trimethylsilyl)-1-propanesulfonic acid-$d_6$ sodium (DSS-$d_6$; Sigma-Aldrich) and was injected in an NMR tube (5 mm outside diameter; Wilmad-Lab Glass). DSS-$d_6$ was used for the calibration of the 0-ppm (parts per million) position and to quantify the product concentrations as an internal standard. A solvent suppression was run to minimize the solvent signal. The NMR peaks were assigned based on the peak amplification by spiking authentic standards and comparison with the spectra of standards.

## Thermodynamic estimation of electrochemical potential that is generated under the icy planetesimal conditions

The redox potentials that were generated by oxidation of $H_2$ as a function of temperature and pH (in Fig. 2a) was calculated by assuming the following half-reaction:

$$2H^+ + 2e^- \rightarrow H2 \tag{3}$$

using the equation

$$E_h = \frac{1}{2F}\left(2RTln\alpha_{H^+} - RTln\alpha_{H_2} - \triangle_f G^0(H_2)\right) \tag{4}$$

In the above equation, $T$, $R$, and $F$ stand for temperature in kelvin, the gas constant (8.31447 J mol⁻¹K⁻¹), and the Faraday constant (96485 C mol⁻¹), respectively. $\alpha_i$ represents the activity of the species i. $\Delta_f G^0(i)$ represents the standard partial molar Gibbs free energy of formation of the species $i$ at desired temperature and pressure, which were calculated according to the revised Helgeson–Kirkham–Flowers (HKF) equations of state[101] together with the thermodynamic data and the revised HKF parameters reported by Shock et al. for $H_2$[102]. In all calculations, the pressure was set to 100 bar. [$H_2$] = 1 mmol kg⁻¹.

## Data availability

The authors declare that the data supporting the findings of this study are available within the paper and its Supplementary Information file. The experimental setup, the HPLC chromatograms, and NMR spectra data of product analyses are provided in the Supplementary Information.

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

## Acknowledgements

This work was funded by Japan Society for the Promotion of Science KAKENHI grant JP20H04608 (Y.L.), JP19K15671 (Y.L.), JP22H05153 (N.K.), JP17H06456 (Y.S.), and JP21K13976 (H.K.). We thank Prof. Naohiro Yoshida for the use of NMR for analysis of organic compounds, Prof. Ryuhei Nakamura and Dr. Daoping He for the support on the electrochemical measurements, Dr. Ruiqin Yi and Ms. Kumiko Nishiuchi on the product analysis, Dr. Shawn McGlynn, and Dr. Kosuke Fujishima for discussion on amino acid chemistry. Yuko Nakano was partially supported by the Assistant Staffing Program by the Gender Equality Section, Diversity Promotion Office, Tokyo Institute of Technology.

## Author contributions

Y.L. and Y.S. conceptualized the study; Y.L., N.K., and Y.S. designed the experiments; N.K. did the thermodynamic calculation; Y.L. and Y.N. conducted the experiments; Y.L., Y.S., H.K. cowrote the paper; K.J.-F. contributed to the discussion and interpretation on the results and edited the manuscript; all authors contributed to the reviewing and editing.

## Competing interests

The authors declare no competing interests.
