## [Peer Review File · Nature Communications]

Geoelectrochemistry-driven alteration of amino acids to derivative organics in carbonaceous chondrite parent bodiesREVIEWER COMMENTS

Reviewer #1 (Remarks to the Author):

Li et al have submitted experimental work to help address alteration / reactivity of amino acids on planetary bodies. Overall, I find this work to be noteworthy and significant. I have a number of comments below that I believe need to be addressed prior to publication.

In the introduction, Strecker and Formose reactions are specifically called out. Reductive amination has also been heavily studied in aqueous environments, as studied by the Moran and Barge groups.

Was this reaction tested in the absence of mineral? spectra from these tests should be included in the SI and noted in the methodology.

The discussion and proposed reaction cycle is very interesting, however this section should connect better to their claim about the importance of H₂.

In the results, it is discussed that the authors used a mixture of racemic L and D enantiomers for the chiral amino acids. However, in the methods, it appears as though they purchased L and D AA's separately. It is extremely difficult to measure out and make a racemic mixture by combining L + D amino acids separately. The authors should comment on this choice.

The supplemental information is well organized and includes significant data. However for the NMR spectra, labels for Gly and Ala are missing for S8 and S9. Also, including comparisons to the standards / the spiked controls used to identify the peaks would be a nice addition to the SI.

Minor notes:

line 318 appears to be highlighted, please remove this.

Fig 4 has a lot of information on the proposed mechanisms but is very busy. It might be beneficial to have one figure for the mechanisms and one for the data, to make this easier to understand.

Reviewer #2 (Remarks to the Author):

The present manuscript by Li et al. reports on experiments that investigate the geoelectrochemical alteration of amino acids in order to understand meteoritic amino acid distributions. I greatly enjoyed reading this manuscript and think that it provides important insights into an unexplored area. In general, I think the manuscript is extremely well written and could be published as is. I have a few questions/suggestions that might help the authors make the manuscript better that I would like for them to consider.

#1 The review uses "free" amino acids (i.e., not Strecker precursors such as amino nitriles or amino amides) as the starting points. Presumably Strecker synthesis, including hydrolysis steps to the free amino acids are occurring at the same time and in the same milieu as the geoelectrochemical alteration. How would the proposed reaction pathways change if they were enacted upon the amino nitrile or amino amide form?

#2 The manuscript briefly discusses the predominance of non-alpha amino acids in the more altered meteorites. I would encourage the authors to comment on a specific hypothesis for the observed tendency of thermally altered meteorites to have higher abundances of n-omega-amino acids (where the amine group is on the carbon farthest from the carboxylic acid; e.g. Burton et al. *Meteoritics & Planetary Science* 2012). If the starting amino acid were a dicarboxylic acid such as aspartic acid or glutamic acid, would the geoelectrochemical alteration lead to decarboxylation of the isolated carboxylate or at alteration of the carboxylate/alpha amine region?

I thank the authors for the interesting manuscript and look forward to seeing the community's response.

Reviewer #3 (Remarks to the Author):

Manuscript #: NCOMMS-21-49841

MANUSCRIPT TITLE: Geoelectrochemistry-driven alteration of amino acids to derivative organics in carbonaceous chondrite parent bodies

JOURNAL SUBMITTED TO: Nature Communications

MANUSCRIPT SUMMARY

This manuscript hypothesizes that low-temperature, geo-electrochemical reactions alter amino acids in water-rock differentiated parent bodies (i.e., icy planetesimals) of carbonaceous chondrites due to pH and redox gradients. This hypothesis was evaluated by performing electrolysis experiments of amino acids at ambient temperature (25 degC) in an electrochemical cell made of two chambers separated by a proton-exchange membrane. This experimental setup was designed to simulate electrochemical conditions within icy planetesimals. Experiments were conducted for 14 days prior to collecting and analyzing samples for conversion products of amino acids, namely primary amines, alpha-hydroxy acids, and monocarboxylic acids. This study used a combination of ion exchange chromatography, HPLC with electric conductivity detection, HPLC-FD, HPLC-UVD, and NMR to analyze the reaction products. The manuscript reports that amino acids were electrochemically altered to monoamines and alpha-hydroxy acids by these experiments. The manuscript further indicates that H₂ can be an important driver for organic evolution in water-rock differentiated CC parent bodies as well as the Solar System icy bodies that would possess similar pH and redox gradients.

STRENGTHS

Major

- 1) This manuscript proposes a unique idea for why a large discrepancy exists between experimental results and meteoritic records regarding the study of organics in small solar system bodies, namely the geo-electrical modification of amino acids into amines, hydroxy acids, and monocarboxylic acids.
- 2) The manuscript provides a large data set to demonstrate how starting reagents of glycine, alanine, and valine precede a variety of amines, monocarboxylic acids, and alpha-hydroxy acids.

3) This manuscript proposes a very interesting idea for why heavily aqueously altered CCs are relatively depleted in amino acids, but relatively rich in monoamines and hydroxy acids – that is, the electrolysis of amino acids in icy planetesimals.

WEAKNESSES

Major

1) The manuscript contained minimal data from the analyses of blanks or controls. The only such data that were found to be reported in either the main text or supplementary material were the mentioning of detecting $9.7 \pm 0.4 \mu\text{M}$ formate and $7.2 \pm 0.5 \mu\text{M}$ acetate contaminants in the 14-day electrolysis experiments (lines 127-128 of the main text) and that methanol was detected in a control sample (Fig. S8 legend). It should be noted that lines 125-126 of the main text stated that amine and alpha-hydroxy acid contaminants were not detected in the starting materials or control experiments, but this was not verified by showing the reader the data. When performing an investigation that incorporates analytical chemistry and organic compounds that are common terrestrial contaminants, under conditions designed to simulate primitive solar system environments, it is imperative to show the data from the analyses of appropriate blanks and controls. It is insufficient to merely tell the reader via text what the results of these analyses were because in such a circumstance, the reader is unable to verify if the text in the manuscript is consistent with the data. Therefore, the reader cannot objectively confirm that the products identified in the experimental samples were not also present in the blanks and controls.

2) English language challenges were a consistent issue throughout the manuscript. The manuscript did not contain individual instances of major lapses in English comprehension, which was greatly appreciated. However, the manuscript did include far too many lesser examples of questionable English language use (in particular, pluralization and tense agreement issues) that cumulatively pose a challenge to readers of this material, making the material cumbersome to read. This issue must be resolved before the manuscript is considered acceptable for publication. Some of the issues have been pointed out below, for convenience. It appears, based on the Deep Carbon Observatory Data Portal, that at least one member of the author list is confirmed to have U.S. nationality and received a Ph.D. at a U.S.-based institution, which would suggest this author is likely to be fluent in English. If so, at least this particular author should thoroughly read the manuscript for English language inaccuracies before submitting this manuscript in the future.

Main Text:

1. Line 41: “meteoritic” not “meteortic”
2. Line 43: “such a tendency” or “such a pattern” not “such tendency”

3. Line 51: "in the presence" not "in presence".
4. Line 94: "acids" not "acid" (this study focuses on the alteration of more than one amino acid)
5. Line 127: "14-day electrolysis experiments" or "14 days of electrolysis" not "14-days electrolysis"
6. Line 138: "Chromatograms" not "Chromatographs" (a chromatograph is an instrument; a chromatogram is the data product of the chromatograph and this figure legend is displaying the data products of the chromatograph, not the chromatograph instrument, itself)
7. Line 139: "primary amine products" not "primary amines products" (this is a superfluous pluralization)
8. Line 140: "Chromatograms" not "Chromatographs"
9. Line 141: "organic acid products" not "organic acids products"
10. Line 143: "chromatograms" not "chromatographs"
11. Line 154: "tended" not "tend"
12. Line 162: is it possible that "active" was intended to be "actively"?
13. Line 267: is it possible that "were" was intended to be "are"? (are the origins of CCs' amines no longer controversial?)
14. Line 290: "protocompartment" not "protocompartments"
15. Lines 301-302: "α-hydroxy acids didn't show consistent homolog tendencies" not "α-hydroxy acids didn't show consistent homolog tendency".
16. Line 310-311: "aqueous phase of a CC's parent body; however," not "aqueous phase in CCs' parent body, however," (see the following website for how to properly use punctuation before the conjunctive adverb "however": <https://www.grammarerrors.com/punctuation/commas-with-conjunctive-adverbs-however-furthermore-etc/>)
17. Line 393: more correct to write "In regard to" as opposed to "As regards" (see the following website for an explanation: <https://www.grammarbook.com/homonyms/in-regards-to-with-regards-to.asp>)
18. Line 396: "dominant" not "dominating"
19. Line 399: "dominant" not "dominating"
20. Line 426: "experiments (18)" not "experiment (18)" (the cited literature involved performing more than one experiment).
21. Line 428: "; however," not ", however," (see explanation in 16, above)
22. Line 497: "blanks" not "blank"

Supplementary Materials:

1. The legend for Figure S8 contains a double negative: "in a control where no amino acid was not implemented." It is better to write this as "in a control where amino acids were not implemented."

3) There were numerous examples where the text, figures, or tables were not consistent with one another. It is critical that the text, figures, and tables are consistent with one another before this manuscript can be considered acceptable for publication. Examples are below:

Main Text:

A. Lines 103-111 reads: "Using these two sulfides as catalysts, the conversion of amino acids to various products, including primary amines, α -hydroxy acids, and monocarboxylic acids was observed after 14-day electrolysis of amino acids at $-0.6\sim-1.0$ V versus standard hydrogen electrode (vs. SHE). We simulated the electrochemical potential range generated at a pH range of $9\sim 13$ and temperature range of $0\sim 150$ oC in the icy planetesimal core (highlighted in blue in Fig. 2A), where H₂ oxidation was redox coupled with the reductive alteration of amino acids."

When evaluating Fig. 2A, the region highlighted in blue spans a pH range of 9-13 and a temperature range of 0.1 degC to 150 degC, which is consistent with the experimental range outlined in the text. However, the region highlighted in blue spans an electric potential range of -0.5 V to -1.0 V, which is not consistent with the -0.6 V to -1.0 V range stipulated in the text.

B. Lines 166-167 read: "Moreover, methylamine and glycolate were generated on both FeS and NiS under a diluted Gly concentration ($0.2\sim 2$ mM) (fig. S14) ."

Upon looking at Fig. S14, glycolic acid concentration is written on the y-axis in the lower pane of Fig. S14. It is likely that this was an honest typo mistake, but glycolate is not glycolic acid, causing Fig. S14 to not agree with what the text stated Fig. S14 showed. The same inconsistency is observed when comparing the text in the header for Table S8 and the text in Table S8.

C. Figure 2C shows that when using NiS as the catalyst, an electrical potential of -0.9 V, and a phosphate (pH 7) buffer, acetate was generated from glycine. When looking at Table S1, the concentration of acetate produced under these conditions was 17 μ M. However, when looking at Fig. S15, the existence of acetate production from glycine when using a phosphate buffer (pH = 7), NiS as a catalyst, and an electrical potential of -0.9 V, is non-existent. Why do the data shown in Fig. 2C and Table S1 appear to disagree with the data shown in Fig. S15?

D. The data that is visually represented in Fig. 5 do not necessarily agree with the same data that is quantitatively represented in Table S1. For example, when using FeS as a catalyst and an electrical potential of -0.8 V, the amount of methylamine derived from glycine is ~ 0.035 mM, according to Fig. 5A. However, the quantity reported for these same conditions is 0.023 mM, according to Table S1. Also, when using NiS as a catalyst and an electrical potential of -1.0 V, the amount of methylamine derived from glycine is ~ 0.016 mM, according to Fig. 5A. However, the quantity reported for these same conditions is 0.0094 mM, according to Table S1. Why do the data shown in Fig. 5A disagree with what is supposed to be the exact same data reported in Table S1?

4) Some of the primary conclusions and implications of this work are not sufficiently supported by evidence. Examples are provided below.

a) The legend for Fig. 4C states that reaction “Steps denoted in dashed lines and arrows were based on speculation and [a] literature report (57).” (Lines 230-231). A literature report is appropriate to use as evidence to support proposed reaction pathways, but speculation is not.

b) Lines 372-387 make the argument that the hydroxy acid/amine ratio is dictated by reduction potential and the identity of the catalyst, and that the reduction potential is tied to the water-rock ratio parameters, and thus the hydroxy acid/amine ratio is tied to water-rock interaction conditions. However, this argument is largely evidenced by an incomplete data set in Fig. 7. For example, there are numerous data points in Fig. 7 that are missing, making it difficult to properly evaluate such a relationship. Furthermore, the comparisons made in Fig. 7 were based on 2 heavily altered CRs. This is an exceedingly small sample set upon which to base such relational conclusions. Perhaps consider evaluating the trends using an $n > 2$ before making conclusions that the trends are likely to be valid.

c) Lines 394-399 make the argument that because the LA/EA ratio in CR2.4 MIL 090001 is similar to that obtained at -0.7 V and -0.8 V on NiS in the electrolysis experiments performed in this work, such a finding “suggests a relatively alkaline pH in the parent planetesimal of MIL chondrites.”

The above conclusion was made based on an individual observation of a ratio similarity between the electrolysis experiments and a singular MIL chondrite. It is not appropriate to extract such sweeping implications for all Miller Range chondrites from just one pair of data points that resulted from using a singular Miller Range chondrite in the comparison.

d) Lines 462-467 proposes that geo-electrochemical alteration of organic molecules (e.g., amino acids) served as a source of ammonia in the interlayers of saponites on Ceres.

This proposal is made without providing any appropriate evidence to support it. For example, amino acids have not yet been reported to have been detected on Ceres material. Furthermore, it is not clear if electrolysis conditions, like those tested in this study, readily occur under Ceres-like conditions to facilitate such ammonia production. It is not appropriate to make such broad-scale proposals without the necessary evidence.

e) Lines 468-471 state the implication that because the organic molecules in Enceladus’ plume possess similar characteristics to those produced by the electrolysis experiments performed in this work, “this implies the presence of amino acids in the subsurface ocean” of Enceladus if, in fact, the plume organics were products of geo-electrochemical reactions at the interface of Enceladus’ rocky core.

This implication was made without appropriate substantiating evidence because it was based on a finite amount of non-highly specific chemical data from observations of plume constituents, while also relying on an unverified prevailing notion (i.e., that geo-electrochemical reactions like those studied in this work occur readily at the interface of Enceladus' core). This implication comes across as rampant speculation, which is not appropriate to include in scientific research literature.

Minor

1) Additional details are needed in the methods section to enable the reader to reproduce the work described in the manuscript.

a. Line 479 states: "conducted in a glove box filled with Ar gas, with 3.97 % H₂ being added (the COY system)."

What is the percentage bases of H₂ being added? Was this a mole percentage basis, a mass percentage basis, a volume percentage basis, etc.? Furthermore, The acronym "COY" was not defined for the reader.

b. Line 495 states that 20 mM of each amino acid was used, unless otherwise noted. Why was this concentration used for amino acids? Is there evidence to cite as justification that 20 mM of a singular amino acid in a 100 mM phosphate buffer (pH 7) solution is representative of a plausible icy planetesimal environment?

c. Lines 496-498 state: "Argon gas was purged into the solution (20 ml min⁻¹) for at least 1 hour prior to the electrolysis. After purging, the solutions were sampled as blank. While keeping the Ar purging, a constant potential was applied on the working electrode by using a multi-channel potentiostat (PS-08; Toho Technical Research)."

Presumably, the need to purge the system with Ar was similar for why the catalysts were prepared in a glove box filled with Ar, which was to prevent oxidation by atmospheric oxygen, as the presence of molecular oxygen can alter the composition of the reaction products. If so, how did collection of the blank occur without breaking the seal and allowing oxygen to enter the reaction apparatus? More generally, the reader needs to know the specific steps used to execute the sampling process that was implemented, so that the reader can properly reproduce the work detailed in this manuscript.

d. Line 501: an explanation for equation (2), or a citation to refer readers to that might otherwise explain equation (2), would be helpful for readers to better understand the context of equation (2). As a side note, there is no equation (1) in the is manuscript, but the manuscript does include equations (2), (3), and (4). The numbering scheme for equations need to start with (1).

2) Many of the figures contained graphics that were of poor resolution or challenging to interpret. Examples are listed below:

a. The “light gray dots” in Fig. 2B are too small and too difficult to see. They need to be a brighter or darker color.

b. The NMR spectra in Figures 4, S8-S13 are of poor quality and must be improved, particularly to allow the reader to better see the small peaks that were pointed out in these figures.

c. The data points in Figure 5 are often so tightly clustered together and of similar colors, making it very difficult to follow the data point trends throughout the subfigures. A combination of color and symbol variations are recommended for use in this figure, as opposed to using the same symbol for each data point and just changing the colors.

3) Lines 180-182 indicated that HPLC concentration measurement errors were <2%; however, none of the quantitative data contains uncertainty estimates. Even though the HPLC measurement error was small, it is nonetheless imperative to include measurement errors when displaying quantitative data in figures and tables. In particular, Figures 3, 5-7, S3-S5, S14-S15, S17-S18, and Tables S1-S3 and S5-S8 need uncertainty estimates to accompany the measurement estimates.

4) Figure S16 shows an NMR spectrum of reaction products with notable analytes pointed out. However, the reaction product NMR spectrum is not accompanied with appropriate standard NMR spectra. The standard spectra are necessary for readers to properly evaluate the veracity of the claims of analyte identification in the reaction product NMR spectrum.

5) The manuscript often refers to GRO 95577 as a CR 2.0, and cites Aponte et al. (2020), MAPS, 55, 2422-2439 (e.g., see Table S4 of the manuscript being reviewed). While it is true that Table 1 of Aponte et al. (2020) states that GRO 95577 is a CR 2.0, but in doing so, the Aponte et al. (2020) article cites the GRO 95577 data from Glavin et al. (2010) MAPS, 45, 1948-1972, and Glavin et al. (2010) states that GRO 95577 is a CR1. Furthermore, the Meteoritical Bulletin Database states that GRO 95577 is a CR1 (<https://www.lpi.usra.edu/meteor/metbull.php?sea=GRO+95577&sfor=names&ants=&nwas=&falls=&valids=&stype=contains&lrec=50&map=ge&browse=&country=All&srt=name&categ=All&mblist=All&rect=&phot=&strewn=&snew=0&pnt=Normal table&code=11305>). Please confirm the meteorite classifications used in the manuscript.

6) The reference to Fig. 3B in Line 194 is incorrect. This needs to be a reference to Fig. 4B.

7) A reference is needed to substantiate the statement made in Lines 263-264.

8) Lines 53-54 state the hypothesis of this work: “Here, we hypothesize that low-temperature geo-electrochemical reactions altered amino acids in water-rock differentiated CCs’ parent bodies (icy planetesimals) due to pH and redox gradients.”

The manuscript did not sufficiently provide evidence that the “pH and redox ‘gradients’” component of the hypothesis was empirically evaluated and proven via experiments in this work. The experimental design incorporated the use of a pH 7 phosphate buffer and applied “a constant potential” (Line 497) to the working electrode. The manuscript stated that the electrochemical potential range of -0.6 to -1.0 V that is generated from a pH range of 9-13 and a temperature range of 0-150 degC in the icy planetesimal was simulated. Therefore, the manuscript is claiming that because a specific electrochemical potential range was simulated, then this means that a pH range of 9-13 was also simulated. This appeared to be the basis for the claim that pH and redox gradients were tested in these electrolysis experiments. However, the electrolysis experiments didn’t implement any pH or electrical potential gradients. Instead, the experiments entailed the use of constant electrical potentials applied in a constant pH environment. The manuscript needs to better substantiate why individually testing constant electrical potentials in a constant pH environment is equivalent to pH and redox gradients.

9) The results when using variable concentrations of starting glycine reagent are not intuitively understood and were not sufficiently evaluated:

a. Figure S14A and S18 showed that the amount of methylamine produced from glycine at -0.9 V drops off significantly from 2 mM glycine to 20 mM glycine. This result is true whether NiS is the catalyst or FeS is the catalyst. Why would the methylamine production be so much smaller from 20 mM glycine than 2 mM glycine? Wouldn’t having a greater amount of starting material facilitate the production of a greater amount of product? This unexpected result needs to be clarified. Furthermore, this trend is notably contradictory to that observed for glycolate production vs glycine concentrations.

RECOMMENDATION

Reject. Explanation for recommendation provided below.

This manuscript has the potential to be of interest to the meteoritic community, particularly the component of this community that actively studies soluble organics. However, the scope of this work comes across as too narrow and specific to be of broad appeal to the greater scientific community. Based on the weaknesses described above, this manuscript does not appear appropriate for publication in a journal with an impact factor of ~15 (i.e., Nature Communications).

There is substantial work that needs to be done to improve this manuscript. However, after improvements have been made, it is possible this work could be suitable for publication in a journal that serves the specific community of interest. Example target journals include Meteoritics & Planetary Science, Geochimica Cosmochimica Acta, or Earth and Planetary Science Letters, to name a few.

Hopefully the authors will take these recommendations under consideration because the basic tenants of the study are interesting and worthy of consideration for publication, but in a different venue than Nature Communications, and after significant improvements have been made to the manuscript.

Response to reviewers

Geoelectrochemistry-driven alteration of amino acids to derivative organics in
carbonaceous chondrite parent bodies (NCOMMS-21-49841)

Yamei Li, Norio Kitadai, Yasuhito Sekine, Hiroyuki Kurokawa, Yuko Nakano, and Kristin
Johnson-Finn

We thank the reviewers and editors for providing constructive comments. In the revised version of our manuscript, we addressed all of the points raised by the reviewers and added new experiments and analyses to further complement our work. Consequently, we showed that our conclusions still hold. We highlighted the revision in blue in the revised manuscript and supplementary document. We appreciate the positive comments highlighting the potential scientific contribution of our study.

Please see below for a point-by-point response to the reviewers' comments.

REVIEWER COMMENTS

Reviewer #1 (Remarks to the Author):

Li et al have submitted experimental work to help address alteration / reactivity of amino acids on planetary bodies. Overall, I find this work to be noteworthy and significant. I have a number of comments below that I believe need to be addressed prior to publication.

Q1: In the introduction, Strecker and Formose reactions are specifically called out. Reductive amination has also been heavily studied in aqueous environments, as studied by the Moran and Barge groups.

A1: Thanks for the comment. We were indeed aware of the reductive amination as an alternative pathway for the synthesis of amino acids and hydroxy acids as reported by Barge group. In the original manuscript, we have cited this research as Ref. 70 (numbered 8 in the revised manuscript). In the revised manuscript, we further added the paper by Moran group as a new reference numbered 9, and described these two pathways in the introduction part (line 40). This pathway relies on α -keto acids as the precursors, however, the availability of α -keto acids in carbonaceous meteorites is largely unclear. To our knowledge, only one study (Cooper, et al., Ref. 75) reported the presence of pyruvate and α -keto glutaric acid in Murchison. The limited chemical diversity of these keto acids cannot be relied on to generate the corresponding amino acid and hydroxy acid counterparts that show much greater isomer

and analog diversities in carbonaceous chondrites. Therefore, we consider this process has limited impact on the amino acid abundances and softened our argument.

We revised the sentences accordingly:

Lines 38~40: "Many aqueous syntheses of amino acids have been demonstrated through either Strecker-(5, 6) or Formose-type (7) reactions" was revised to "Many aqueous syntheses of amino acids have been demonstrated through either Strecker-(5, 6) or Formose-type (7), and reductive amination (8, 9) reactions"

Line 297: "reduction of α -keto acids by Fe hydroxides (70)" was revised to "reduction of α -keto acids by Fe hydroxides (8, 9)."

Lines 297~298: "Nevertheless, the discovery of only one α -keto acids (pyruvate) in meteorites (72) suggests that reduction of α -keto acid cannot generate all the α -hydroxy acids with comparable isomer and analog varieties in CCs." was revised to "Nevertheless, the discovery of only two α -keto acids (pyruvate and α -keto glutaric acid) in meteorites (75) suggests that reduction of α -keto acid cannot provide a reliable pathway to generate all the α -hydroxy acids with comparable isomer and analog varieties in CCs."

Q2: Was this reaction tested in the absence of mineral? spectra from these tests should be included in the SI and noted in the methodology.

A2: Thanks for the suggestion. We have conducted the control experiment to test glycine electrolysis in the absence of minerals and described the result in the original manuscript (Line 164~166 in the original manuscript; Line 173~175 in the revised manuscript). The statement is as follows:

"A control experiment without a metal sulfide catalyst does not generate methylamine and glycolate from Gly, suggesting the key role of sulfide catalysts."

In the revised manuscript, the spectra of the sample generated from this experiment have been added in the supplementary Fig. S21 and referred to in the associated text (Line 174). The method of collecting this control sample has been described in the methodology as follows:

Lines 588~591: A control experiment without using a metal sulfide catalyst was conducted by two-week electrolysis of Gly (20 mM) in a phosphate buffered electrolyte

(pH 7) at -0.8 V using a bare carbon paper electrode. Another control experiment was conducted by two-week electrolysis of a phosphate buffered electrolyte (pH 7) free of amino acids at -0.9 V using FeS as a catalyst.

Q3: The discussion and proposed reaction cycle is very interesting; however, this section should connect better to their claim about the importance of H₂.

A3: Thanks for the suggestion. We have made several revisions on Figure 4 to make it clearer. Firstly, we separated the NMR data from the reaction mechanism and made two individual figures (reaction mechanisms in Figure 4; ¹H-NMR spectra in Figure 5). Secondly, to show the importance of H₂ in the overall mechanism, we have highlighted the steps of electron generation from H₂ oxidation reaction and the participation of electrons in the reaction steps as shown in Figure 4. The figure captions were revised accordingly.

Q4: In the results, it is discussed that the authors used a mixture of racemic L and D enantiomers for the chiral amino acids. However, in the methods, it appears as though they purchased L and D AA's separately. It is extremely difficult to measure out and make a racemic mixture by combining L + D amino acids separately. The authors should comment on this choice.

A4: Thanks for raising the point. Racemic amino acids were prepared by mixing the corresponding D- and L-amino acids in a 50:50 ratio (mol%). We have confirmed that the concentration of each enantiomer was as we desired by using a UV-HPLC system with a chiral column.

Q5: The supplemental information is well organized and includes significant data. However for the NMR spectra, labels for Gly and Ala are missing for S8 and S9. Also, including comparisons to the standards / the spiked controls used to identify the peaks would be a nice addition to the SI.

A5: Thanks for raising this problem. The NMR peaks for Gly, Ala, and Val in spectra S8~S13 have been labeled. In addition, the NMR peaks for Ala have been labeled in Figure 5B. The spectra of standard compounds were also collected and added in each NMR spectra figures (Supplementary Figs S8~S13, Fig. S16).

Minor notes:

Q6: line 318 appears to be highlighted, please remove this.

A6: The highlight has been removed.

Q7: Fig 4 has a lot of information on the proposed mechanisms but is very busy. It might be beneficial to have one figure for the mechanisms and one for the data, to make this easier to understand.

A7: Thanks for the constructive suggestion. The NMR spectral data (original Figs. 4B and 4D) have been removed from the original Figure 4, and depicted as a separate figure (Fig. 5).

Reviewer #2 (Remarks to the Author):

The present manuscript by Li et al. reports on experiments that investigate the geoelectrochemical alteration of amino acids in order to understand meteoritic amino acid distributions. I greatly enjoyed reading this manuscript and think that it provides important insights into an unexplored area. In general, I think the manuscript is extremely well written and could be published as is. I have a few questions/suggestions that might help the authors make the manuscript better that I would like for them to consider.

Reply: we thank the reviewer for the thoughtful review and positive comments on our study.

Q8: The review uses "free" amino acids (i.e., not Strecker precursors such as amino nitriles or amino amides) as the starting points. Presumably Strecker synthesis, including hydrolysis steps to the free amino acids are occurring at the same time and in the same milieu as the geoelectrochemical alteration. How would the proposed reaction pathways change if they were enacted upon the amino nitrile or amino amide form?

A8: Thanks for the inspiring comments. We have conducted additional experiments with aminoacetonitrile as the starting compound using FeS and NiS as catalysts. The experiments were operated at two potentials (-0.6V and -0.9V) at a starting aminoacetonitrile concentration of 20 mM. After two-week electrolysis at -0.9V, aminoacetonitrile completely converted to various products based on ¹H-NMR spectra (Fig. R3). Under a relatively mild potential (-0.6V), aminoacetonitrile was partly consumed. The IC and fluorescence chromatograms were shown in Figure R1 and R2, respectively. A small Gly peak at 3.88 ppm was resolved in ¹H-NMR spectra under all the conditions, with an estimated concentration of 82~902 μM. Gly was also identified using HPLC together with methylamine and β-alanine (Fig. R2). Therefore, the result suggests that aminoacetonitrile was partially converted to amino acids (Gly in this case) at pH 7 by electrochemistry. Methylamine was detected under all the four conditions, where glycine electrolysis serves as one of the possible pathways. The absence of glycolate as a Gly-mediated product may be caused by 1) the reaction of

aminoacetonitrile or other products competes with the electrolysis of Gly. 2) the in situ formed Gly concentration is too low to compete with more reactive aminoacetonitrile. The multiple peaks in both NMR and fluorescence chromatograms suggest that many amino-bearing compounds were generated and the reaction network of aminoacetonitrile electrolysis is rather complex. After comparing those peaks with 19 types of α -amino acid standards, it was concluded that except for Gly, none of those proteinogenic α -amino acids were generated. The identification of these products requires more sophisticated analytic methods with derivatization method (mass spectroscopy) which is currently not readily available in our group. Since the exact mechanism is beyond the scope of this study, we will target this reaction as a future work. We thank the reviewer for this inspiring comment.

Fig. R1 IC chromatograms of sample products generated by electrolysis of aminoacetonitrile. The bottom one shows the chromatogram of standard compounds. After electrolysis, formate and acetate were formed as the major carboxylate products.

Fig. R2 Fluorescence chromatograms of sample products generated by electrolysis of aminoacetonitrile. The bottom one shows the chromatogram of standard compounds. After electrolysis, glycine, β -alanine, and methylamine were formed as the major amino-bearing products. Other peaks are not assigned due to lack of standards.

Fig. R3 $^1\text{H-NMR}$ spectra of sample products generated by electrolysis of aminoacetonitrile. After electrolysis at -0.9V for two weeks, aminoacetonitrile was almost completely consumed. Glycine, formate and acetate were detected, while other products are not assigned due to lack of standards.

Q9: The manuscript briefly discusses the predominance of non-alpha amino acids in the more altered meteorites. I would encourage the authors to comment on a specific hypothesis for the observed tendency of thermally altered meteorites to have higher abundances of n-omega-amino acids (where the amine group is on the carbon farthest from the carboxylic acid; e.g. Burton et al. *Meteoritics & Planetary Science* 2012). If the starting amino acid were a dicarboxylic acid such as aspartic acid or glutamic acid, would the geoelectrochemical alteration lead to decarboxylation of the isolated carboxylate or at alteration of the carboxylate/alpha amine region?

A9: Thanks for the thoughtful comment.

1) Comment on a hypothesis for the observed tendency of thermally altered meteorites to be predominated by straight chain, amino terminal (n- ω -amino) acids.

In 2012, Burton et al. reported that thermally altered meteorites, including CO3, CV3, and Antarctic ureilites are dominated by straight chain, amino terminal (n- ω -amino) acids. Later, similar tendency was observed on thermally metamorphosed R and CK chondrites (Burton

et al., 2014). The range of temperatures that experienced by these meteorites are estimated to be 250–600 °C (CO and CV) or 300–600 °C or warmer (R and CK). In their papers, they argued that this may be due to the dominating effect of Fischer–Tropsch-type (FFT) synthesis. However, this argument faces challenges that 1) the very low amino acid abundances in specific CC types (CK and R chondrites) cannot be well explained; 2) experimental FFT syntheses have not been able to reproduce the similar organic distribution.

Based on our model and experimental results, in addition to the synthetic pathway, the decomposition process should be carefully considered and included to account for the meteoritic amino acid distribution. The higher thermal stability of n- ω -amino acids have been reported by Islam, et al. in 2003. These n- ω -amino acids were found to form intramolecular lactam condensates via dehydration, thus showing high thermal stability. In addition, repositioning the NH₂ group from the α -carbon to the terminal carbon reduces inductive and electrostatic effects favoring decarboxylation (Li, et al., International Journal of Chemical Kinetics, 2003, 35(11): 602-610). Therefore, one possibility is that the elevated relative abundances of n- ω -amino acids are not due to their preferential syntheses but that other isomers are more rapidly destroyed due to the inability to form stable lactam like products. The thermal decomposition kinetics would rely on the temperature, mineral assemblages, and redox conditions. To testify the mechanism, the possible decomposition products can be targeted and analyzed. Aqueous alteration of amino acids would generate amines, hydroxy acids, and monocarboxylic acids. By contrast, thermal decomposition usually decomposes the amino acids into N₂, CO₂, and H₂O (as reported in I.M.Weiss, et al., BMC Biophysics, 2018, 11(1), 1-15). This can well explain why the amine abundances on thermally metamorphosed CO, CK chondrites are so low although amino acids were substantially decomposed.

We have added discussion on this matter as shown below.

Lines 474~490:

Additionally, thermally altered meteorites, including CO3, CV3, Antarctic ureilites, R, and CK chondrites, have been reported to be dominated by straight chain, amino terminal (n- ω -amino) acids (79, 80). The range of temperatures that these meteorites have experienced is estimated to be 250–600 °C in CO and CV chondrites and 300–600 °C or warmer in R and CK ones. Although Fischer–Tropsch-type (FFT) synthesis has been argued to account for these phenomena, this argument faces challenges that 1) the very low amino acid abundances in specific CC types (CK and R chondrites) cannot be well explained; 2) experimental FFT syntheses have not been able to reproduce the similar organic distribution. Based on our model and experimental results, in addition to the synthetic pathway, the decomposition process should be carefully considered and included to account for the

meteoritic amino acid distribution. The higher thermal stability of n- ω -amino acids than other isomers has been reported (52), due to the capability of these n- ω -amino acids to form intramolecular lactam condensates via dehydration. In addition, repositioning the NH₂ group from the α -carbon to the terminal carbon reduces inductive and electrostatic effects favoring decarboxylation (18). Thus, the elevated relative abundances of n- ω -amino acids are possibly not mainly due to their preferential syntheses but that other isomers are more rapidly destroyed due to the lack of the ability to form stable lactam like products. In this regard, we believe that the detection of targeted decomposition products can shed further insight into this question (51).

2) Regarding the second comment, we have conducted additional experiments with aspartic acid (Asp, 20 mM) or glutamic acid (Glu, 20 mM) as the starting compounds at -0.9V on both FeS and NiS. The chromatograms of product solutions were shown below for the detection of organic acids and amine-containing products. The data were added as supplementary figures S22~S25 in the revised supplementary document. It was found that γ -aminobutyrate and α -hydroxyglutaric acid were generated from Glu, presumably through decarboxylation and deamination processes. Meanwhile, starting from Asp, β -alanine and malate were also generated via similar pathways. Additionally, glycine, methylamine, ammonia, acetate and formate were detected as common products. This indicate that electrolysis of Asp and Glu also contains reaction pathways mediated via Gly-like intermediate. Taken together, the reaction pathways illustrated in Fig. 4A can be extended to the α -amino acids with anionic side chains. The reaction pathways of electrolysis of Glu and Asp were summarized in Fig. S26. These results also suggest that the decarboxylation occurred in the carboxyl group near the amine region. We have added the discussion on the reaction pathways of these two amino acids with anionic side chains. Additionally, Asp was detected as an unexpected product from Glu electrolysis, where a possible mechanism was proposed.

The added discussion is as follows:

Lines 254~266

To explore the universality of the reaction mechanisms of α -amino acids shown in Fig. 4A, aspartate (Asp) and glutamate (Glu) with anionic side chains were used as the starting substrates (20 mM) for electrolysis at -0.9 V on either FeS or NiS. The chromatograms of product solutions were shown in Fig. S22~S25. Starting from Glu, γ -aminobutyrate and α -hydroxyglutaric acid were generated, presumably through decarboxylation and deamination processes. Meanwhile, starting from Asp, β -alanine and malate were also generated presumably via similar pathways. Additionally, glycine, methylamine, ammonia, acetate and formate were detected as common products. This indicate that electrolysis of Asp and Glu also contains reaction pathways mediated via Gly-like intermediates. Taken together, the

reaction pathways illustrated in Fig. 4A can be extended to the α -amino acids with anionic side chains. The reaction pathways of electrolysis of Glu and Asp were summarized in Fig. S26. Notably, Glu electrolysis also generated Asp and Asp-derived compounds (malate and β -alanine). This suggest another reaction pathway with a C-C bond breaking and recombination of a Gly-like and acetyl group. The exact mechanism is unclear and will be investigated in future.

Based on these results, the reactivity of amino acids with different side chains would be different. Therefore, we plan to further study these reactions in terms of kinetics and mechanisms, in addition to other polar amino acids (e.g., serine and threonine), in more detail in a future project.

I thank the authors for the interesting manuscript and look forward to seeing the community's response.

Reply: we thank the reviewer for the constructive review and inspiring comments on our study.

Reviewer #3 (Remarks to the Author):

Manuscript #: NCOMMS-21-49841

MANUSCRIPT TITLE: Geoelectrochemistry-driven alteration of amino acids to derivative organics in carbonaceous chondrite parent bodies

JOURNAL SUBMITTED TO: Nature Communications

MANUSCRIPT SUMMARY

This manuscript hypothesizes that low-temperature, geo-electrochemical reactions alter amino acids in water-rock differentiated parent bodies (i.e., icy planetesimals) of carbonaceous chondrites due to pH and redox gradients. This hypothesis was evaluated by performing electrolysis experiments of amino acids at ambient temperature (25 degC) in an electrochemical cell made of two chambers separated by a proton-exchange membrane. This experimental setup was designed to simulate electrochemical conditions within icy planetesimals. Experiments were conducted for 14 days prior to collecting and analyzing samples for conversion products of amino acids, namely primary amines, alpha-hydroxy

acids, and monocarboxylic acids. This study used a combination of ion exchange chromatography, HPLC with electric conductivity detection, HPLC-FD, HPLC-UVD, and NMR to analyze the reaction products. The manuscript reports that amino acids were electrochemically altered to monoamines and alpha-hydroxy acids by these experiments. The manuscript further indicates that H₂ can be an important driver for organic evolution in water-rock differentiated CC parent bodies as well as the Solar System icy bodies that would possess similar pH and redox gradients.

STRENGTHS

Major

1) This manuscript proposes a unique idea for why a large discrepancy exists between experimental results and meteoritic records regarding the study of organics in small solar system bodies, namely the geo-electrical modification of amino acids into amines, hydroxy acids, and monocarboxylic acids.

2) The manuscript provides a large data set to demonstrate how starting reagents of glycine, alanine, and valine precede a variety of amines, monocarboxylic acids, and alpha-hydroxy acids.

3) This manuscript proposes a very interesting idea for why heavily aqueously altered CCs are relatively depleted in amino acids, but relatively rich in monoamines and hydroxy acids – that is, the electrolysis of amino acids in icy planetesimals.

Reply: we thank the reviewer for the thoughtful review and positive comments on our study.

WEAKNESSES

Major

Q10: The manuscript contained minimal data from the analyses of blanks or controls. The only such data that were found to be reported in either the main text or supplementary material were the mentioning of detecting $9.7 \pm 0.4 \mu\text{M}$ formate and $7.2 \pm 0.5 \mu\text{M}$ acetate contaminants in the 14-day electrolysis experiments (lines 127-128 of the main text) and that methanol was detected in a control sample (Fig. S8 legend). It should be noted that lines 125-126 of the main text stated that amine and alpha-hydroxy acid contaminants were not

detected in the starting materials or control experiments, but this was not verified by showing the reader the data. When performing an investigation that incorporates analytical chemistry and organic compounds that are common terrestrial contaminants, under conditions designed to simulate primitive solar system environments, it is imperative to show the data from the analyses of appropriate blanks and controls. It is insufficient to merely tell the reader via text what the results of these analyses were because in such a circumstance, the reader is unable to verify if the text in the manuscript is consistent with the data. Therefore, the reader cannot objectively confirm that the products identified in the experimental samples were not also present in the blanks and controls.

A10: Thanks for the useful suggestions. We have added the $^1\text{H-NMR}$ spectra of starting compounds (Gly, Ala, Val, and phosphate buffer) in the supplementary Figure S19. Additionally, we added the product analysis results in a control experiment where no amino acids were implemented in the supplementary Figure S20. The methods to collect these control samples were described in Methods part as follows:

Lines 574~578:

A control experiment without using a metal sulfide catalyst was conducted by two-week electrolysis of Gly (20 mM) in a phosphate buffered electrolyte (pH 7) at -0.8 V using a bare carbon paper electrode. Another control experiment was conducted by two-week electrolysis of a phosphate buffered electrolyte (pH 7) free of amino acids at -0.9 V using FeS as a catalyst.

Q11: English language challenges were a consistent issue throughout the manuscript. The manuscript did not contain individual instances of major lapses in English comprehension, which was greatly appreciated. However, the manuscript did include far too many lesser examples of questionable English language use (in particular, pluralization and tense agreement issues) that cumulatively pose a challenge to readers of this material, making the material cumbersome to read. This issue must be resolved before the manuscript is considered acceptable for publication. Some of the issues have been pointed out below, for convenience. It appears, based on the Deep Carbon Observatory Data Portal, that at least one member of the author list is confirmed to have U.S. nationality and received a Ph.D. at a U.S.-based institution, which would suggest this author is likely to be fluent in English. If so, at least this particular author should thoroughly read the manuscript for English language inaccuracies before submitting this manuscript in the future.

Main Text:

1. Line 41: "meteoritic" not "meteortic"

2. Line 43: “such a tendency” or “such a pattern” not “such tendency”
3. Line 51: “in the presence” not “in presence”.
4. Line 94: “acids” not “acid” (this study focuses on the alteration of more than one amino acid)
5. Line 127: “14-day electrolysis experiments” or “14 days of electrolysis” not “14-days electrolysis”
6. Line 138: “Chromatograms” not “Chromatographs” (a chromatograph is an instrument; a chromatogram is the data product of the chromatograph and this figure legend is displaying the data products of the chromatograph, not the chromatograph instrument, itself)
7. Line 139: “primary amine products” not “primary amines products” (this is a superfluous pluralization)
8. Line 140: “Chromatograms” not “Chromatographs”
9. Line 141: “organic acid products” not “organic acids products”
10. Line 143: “chromatograms” not “chromatographs”
11. Line 154: “tended” not “tend”
12. Line 162: is it possible that “active” was intended to be “actively”?
13. Line 267: is it possible that “were” was intended to be “are”? (are the origins of CCs’ amines no longer controversial?)
14. Line 290: “protocompartment” not “protocompartments”
15. Lines 301-302: “ α -hydroxy acids didn’t show consistent homolog tendencies” not “ α -hydroxy acids didn’t show consistent homolog tendency”.
16. Line 310-311: “aqueous phase of a CC’s parent body; however,” not “aqueous phase in CCs’ parent body, however,” (see the following website for how to properly use punctuation before the conjunctive adverb “however”: <https://www.grammarerrors.com/punctuation/commas-with-conjunctive-adverbs-however-furthermore-etc/>)
17. Line 393: more correct to write “In regard to” as opposed to “As regards” (see the following website for an explanation: <https://www.grammarbook.com/homonyms/in-regards-to-with-regards-to.asp>)
18. Line 396: “dominant” not “dominating”
19. Line 399: “dominant” not “dominating”
20. Line 426: “experiments (18)” not “experiment (18)” (the cited literature involved performing more than one experiment).
21. Line 428: “; however,” not “, however,” (see explanation in 16, above)
22. Line 497: “blanks” not “blank”

A11: Thanks for the careful reviewing on English language issues. We appreciate these

comments and have revised accordingly, as highlighted in blue in the main text. In addition, all the coauthors have carefully read the revised manuscript for editing the English language inaccuracies.

Supplementary Materials:

Q12. The legend for Figure S8 contains a double negative: “in a control where no amino acid was not implemented.” It is better to write this as “in a control where amino acids were not implemented.”

A12: The associated text has been corrected.

Q13 There were numerous examples where the text, figures, or tables were not consistent with one another. It is critical that the text, figures, and tables are consistent with one another before this manuscript can be considered acceptable for publication. Examples are below:

Main Text:

A. Lines 103-111 reads: “Using these two sulfides as catalysts, the conversion of amino acids to various products, including primary amines, α -hydroxy acids, and monocarboxylic acids was observed after 14-day electrolysis of amino acids at $-0.6\sim-1.0$ V versus standard hydrogen electrode (vs. SHE). We simulated the electrochemical potential range generated at a pH range of 9~13 and temperature range of 0~150 oC in the icy planetesimal core (highlighted in blue in Fig. 2A), where H₂ oxidation was redox coupled with the reductive alteration of amino acids.”

When evaluating Fig. 2A, the region highlighted in blue spans a pH range of 9-13 and a temperature range of 0.1 degC to 150 degC, which is consistent with the experimental range outlined in the text. However, the region highlighted in blue spans an electric potential range of -0.5 V to -1.0 V, which is not consistent with the -0.6 V to -1.0 V range stipulated in the text.

A13: Thanks for raising this concern. We agree that electrolysis at -0.5 V should be studied. Therefore, we have conducted new experiments at -0.5V using FeS or NiS as the catalyst. The product analysis data were added in Fig. 3, Fig. 6, and Fig. 8 in the main text of the manuscript and the analytical data were added in Fig. S8~S13 (NMR spectra), Fig. S6 (fluorescence chromatograms) and Fig. S7 (IC chromatograms). Since the derivative products were detected at -0.5V, we revised the first paragraph of the Discussion section.

Lines 283~286:

Before revision:

The threshold potential for activating amino acid alteration is -0.6 V, which corresponds to pH 10.5 at 0.1 °C, pH 9.1 at 50 °C, pH 8.5 at 100 °C or pH 7.9 at 150 °C at an H₂ concentration of 1 mmolkg⁻¹ at 100 bar (Fig. 2A). Alkaline pH and higher temperature

can readily generate potentials more negative than -0.6 V.

After revision:

The threshold potential for activating amino acid alteration is -0.5 V, which corresponds to pH 8.7 at 0.1 °C, pH 7.5 at 50 °C, pH 7.1 at 100 °C or pH 6.7 at 150 °C at an H_2 concentration of 1 mmolkg⁻¹ at 100 bar (Fig. 2A). Alkaline pH and higher temperature can readily generate potentials more negative than -0.5 V.

Q14. Lines 166-167 read: “Moreover, methylamine and glycolate were generated on both FeS and NiS under a diluted Gly concentration (0.2 – 2 mM) (fig. S14) .”

Upon looking at Fig. S14, glycolic acid concentration is written on the y-axis in the lower pane of Fig. S14. It is likely that this was an honest typo mistake, but glycolate is not glycolic acid, causing Fig. S14 to not agree with what the text stated Fig. S14 showed. The same inconsistency is observed when comparing the text in the header for Table S8 and the text in Table S8.

A14: We have revised the typo mistakes in the y-axis labels of Fig. S14 and text in the header for Table S8.

Q15. Figure 2C shows that when using NiS as the catalyst, an electrical potential of -0.9 V, and a phosphate (pH 7) buffer, acetate was generated from glycine. When looking at Table S1, the concentration of acetate produced under these conditions was 17 μM. However, when looking at Fig. S15, the existence of acetate production from glycine when using a phosphate buffer (pH = 7), NiS as a catalyst, and an electrical potential of -0.9 V, is non-existent. Why do the data shown in Fig. 2C and Table S1 appear to disagree with the data shown in Fig. S15?

A15: Thanks for raising this inconsistency. We made a mistake in plotting using FeS-catalyzed data for the phosphate-buffered experiment. We are very sorry for this mistake. We have replotted the Fig. S15 using the correct data.

Q16. The data that is visually represented in Fig. 5 do not necessarily agree with the same data that is quantitatively represented in Table S1. For example, when using FeS as a catalyst and an electrical potential of -0.8 V, the amount of methylamine derived from glycine is ~ 0.035 mM, according to Fig. 5A. However, the quantity reported for these same conditions is 0.023 mM, according to Table S1. Also, when using NiS as a catalyst and an electrical potential of -1.0 V, the amount of methylamine derived from glycine is ~ 0.016 mM, according to Fig. 5A. However, the quantity reported for these same conditions is 0.0094 mM, according to Table

S1. Why do the data shown in Fig. 5A disagree with what is supposed to be the exact same data reported in Table S1?

A16: Thanks for raising this point. The reason why the exhibited data in Fig. 5A is relatively larger than the data listed in the Table S1 is that methylamine is not only generated from Gly electrolysis, but also from the electrolysis of Ala and Val. In the original manuscript, we plotted the total amount of methylamine generated from all the three amino acid, including Gly, Ala, and Val, under the same condition. Therefore, the value is relatively higher than the one obtained from Gly only, as listed in Table S1. We noticed that the method has not been described correctly and clearly to the readers in the previous manuscript. In the revised manuscript, we have added one sentence in the caption of Figure 5 to explain that the presented methylamine concentration in Figure 6A is the sum of products from all amino acids (Gly, Ala, and Val).

Lines 320~322:

Since all these three amino acids generate methylamine, the presented methylamine concentration is the sum of values from all the three amino acid electrolysis experiments operated under the same condition.

Additionally, since both Gly and Val electrolysis generate glycolate, we did similar treatment on the presented glycolate concentration data shown in Fig. 6C and described the method in the Figure caption as below:

Lines 326~328:

Since both Gly and Val generate glycolate upon electrolysis, the presented glycolate concentration is the sum of values from Gly and Val electrolysis experiments operated under the same condition.

Q17 Some of the primary conclusions and implications of this work are not sufficiently supported by evidence. Examples are provided below.

a) The legend for Fig. 4C states that reaction “Steps denoted in dashed lines and arrows were based on speculation and [a] literature report (57).” (Lines 230-231). A literature report is appropriate to use as evidence to support proposed reaction pathways, but speculation is not.

A17: Thanks for the comment. Here, we used the term “speculation” for indicating the chemical species at hypothetical intermediate states, such as CH_x and NH_x intermediates, which are necessary for the generation of methylamine. However, as the reviewer suggested, speculation is not an appropriate term to be used in the scientific paper. Therefore, we replaced “speculation” by a more concrete phrase “hypothetical intermediate states” to avoid confusion (Line 241~242).

Q18 Lines 372-387 make the argument that the hydroxy acid/amine ratio is dictated by reduction potential and the identity of the catalyst, and that the reduction potential is tied to the water-rock ratio parameters, and thus the hydroxy acid/amine ratio is tied to water-rock interaction conditions. However, this argument is largely evidenced by an incomplete data set in Fig. 7. For example, there are numerous data points in Fig. 7 that are missing, making it difficult to properly evaluate such a relationship. Furthermore, the comparisons made in Fig. 7 were based on 2 heavily altered CRs. This is an exceedingly small sample set upon which to base such relational conclusions. Perhaps consider evaluating the trends using an $n > 2$ before making conclusions that the trends are likely to be valid.

A18: Thanks for the constructive comment. We indeed agree with the reviewer that the number of carbonaceous samples were too small; however, the choice of these two heavily altered samples was based on the following reasons: 1) geo-electrochemistry is driven by aqueous alteration, therefore it will majorly affect the aqueously altered samples and likely show little effect on primitive samples with limited aqueous activity; 2) the available data of both the hydroxy acid and monoamine abundances are rather limited for aqueously altered samples; 3) the literature that report the hydroxy acid and monoamine concentrations usually used two different samples, which may cause problem due to sample heterogeneity. These considerations led us choose the two samples in the original manuscript.

However, we indeed agree with the reviewer that more samples should be added to make more reliable discussions despite that the available limited data may not be ideal. Therefore, we have added several other CR chondrite samples in Figure 8C, 8F, and 8I, and found that our main conclusions still hold. Notably, for the two more primitive CR2.7 chondrites (GRA 95229 and MIL 090657), both the MA/GA and EA/LA ratios are much lower than 1 (Table S8 for the values). This suggests little effect of geo-electrochemical alteration on these less altered samples. We have added this new discussion in the text as follows:

Lines 449-454

Notably, for the two more primitive CR2.7 chondrites (GRA 95229 and MIL 090657), both the MA/GA and EA/LA ratios are much lower than 1 (Table S8 for the values). This suggests little effect of geo-electrochemical alteration on these less altered samples. Much higher ratio values can be seen in the most primitive CR2.8 samples (EET 92042 and QUE 99177), albeit little effect of geo-electrochemistry can be anticipated due to the limited aqueous activity.

Q19 Lines 394-399 make the argument that because the LA/EA ratio in CR2.4 MIL 090001 is similar to that obtained at -0.7 V and -0.8 V on NiS in the electrolysis experiments performed in this work, such a finding “suggests a relatively alkaline pH in the parent

planetesimal of MIL chondrites.”

The above conclusion was made based on an individual observation of a ratio similarity between the electrolysis experiments and a singular MIL chondrite. It is not appropriate to extract such sweeping implications for all Miller Range chondrites from just one pair of data points that resulted from using a singular Miller Range chondrite in the comparison.

A19: We agree with the reviewer that the data points are too few to make such discussion on the implications. Therefore, we have deleted the related sentence.

Before revision:

Supposing NiS is the dominate catalyst, this suggests that the T and pH of parent bodies of MIL/CR2.4 would be close to pHs of 11.5~13.3 at 25 °C, 9.9~11.2 at 100 °C, and 9.1~10.3 at 150 °C, to generate the potential of -0.7~-0.8 V (data can be extracted from Fig. 2A). This suggests a relatively alkaline pH in the parent planetesimal of MIL chondrites.

After revision:

Supposing NiS is the dominate catalyst, this suggests that the T and pH of parent bodies of MIL/CR2.4 would be close to pHs of 11.5~13.3 at 25 °C, 9.9~11.2 at 100 °C, and 9.1~10.3 at 150 °C, to generate the potential of -0.7~-0.8 V (data can be extracted from Fig. 2A).

Q20 Lines 462-467 proposes that geo-electrochemical alteration of organic molecules (e.g., amino acids) served as a source of ammonia in the interlayers of saponites on Ceres.

This proposal is made without providing any appropriate evidence to support it. For example, amino acids have not yet been reported to have been detected on Ceres material. Furthermore, it is not clear if electrolysis conditions, like those tested in this study, readily occur under Ceres-like conditions to facilitate such ammonia production. It is not appropriate to make such broad-scale proposals without the necessary evidence.

Q21 Lines 468-471 state the implication that because the organic molecules in Enceladus' plume possess similar characteristics to those produced by the electrolysis experiments performed in this work, “this implies the presence of amino acids in the subsurface ocean” of Enceladus if, in fact, the plume organics were products of geo-electrochemical reactions at the interface of Enceladus' rocky core.

This implication was made without appropriate substantiating evidence because it was based on a finite amount of non-highly specific chemical data from observations of plume constituents, while also relying on an unverified prevailing notion (i.e., that geo-electrochemical reactions like those studied in this work occur readily at the interface of Enceladus' core). This implication comes across as rampant speculation, which is not appropriate to include in scientific research literature.

Reply to Q20 and Q21: Thanks for the critical comment on these points. Our original intention is to provide useful insight or implications to guide our future exploration of extraterrestrial bodies based on our results. We agree with the reviewer that our discussion is specific and not very clear to the readers. To clarify this point, we have revised the sentences as follows: We propose that the geoelectrochemical-alteration of amino acids might be the origin of derivative organics and relevant molecules found on the solar system icy bodies, which can be tested with future in situ explorations. On Ceres, which is analogous to volatile-rich CCs (90), ammoniated saponite is known to be present widespread on the surface (91). The surface materials of Ceres would reflect materials contained in the subsurface ocean at the earlier stage of its evolution (92). The Cassini spacecraft found the presence of complex organic molecules and ammonia in plume materials of Enceladus (32, 93-95). These bodies possibly accreted amino acids when they formed. Comets record the building blocks of the outer solar system bodies and are known to contain amino acids, which are thought to be interstellar in origin (96). Spontaneous generation of amino acids in the interstellar medium is experimentally demonstrated to be possible by irradiation of interstellar ice containing primordial molecules by ultraviolet light (66, 67) or high-energy protons (97). This suggests the possibly wide distribution of amino acids in the outer solar system bodies due to the accretion of pristine ices. Amino acids were typically considered stable under low-temperature conditions (0~150 °C); however, our results suggested that redox gradients generated by water/rock interaction could alter amino acids into various amino- and carbonyl-bearing compounds with small molecular weight (e.g., ammonia, methylamine, ethylamine, formate, acetate) (Table S1~S3). McSween et al. (90) have proposed that Ceres' ammonia could have been derived from the decomposition of organics at high temperatures (~300 °C). Our study provides experimental support for this hypothesis, even at lower temperatures. Thus, we suggest that geo-electrochemical alteration of organic molecules (e.g., amino acids) could serve as a source of ammonia in the interlayers of saponites on Ceres and in the plume of Enceladus. The plume includes low-mass amine and carbonyls (93), which is also consistent with our results of derivative products of amino acids. Provided that there exist redox gradients at their rocky core–water mantle boundaries (31, 77), geo-electrochemical alteration of amino acids, if any, could possibly contribute to the detected organics and ammonia. We propose that further exploration missions for Ceres (98), Enceladus (99), and other icy bodies can test our geo-electrochemistry model. Sampling and analyzing their surface materials would clarify the existence of amino acids and mineral catalysts and testify their relations to other compounds. Additionally, we have softened the claim in the final sentence of the abstract.

Original: Our results thus suggest that H₂ can be an important driver for organic evolution in water-rock differentiated CC parent bodies as well as the Solar System icy bodies that **would** possess similar pH and redox gradients.

Revised: Our results thus suggest that H₂ can be an important driver for organic evolution in water-rock differentiated CC parent bodies as well as the Solar System icy bodies that **might** possess similar pH and redox gradients.

Minor

Q22 Additional details are needed in the methods section to enable the reader to reproduce the work described in the manuscript.

a. Line 479 states: “conducted in a glove box filled with Ar gas, with 3.97 % H₂ being added (the COY system).”

What is the percentage bases of H₂ being added? Was this a mole percentage basis, a mass percentage basis, a volume percentage basis, etc.? Furthermore, The acronym “COY” was not defined for the reader.

A22: The percentage bases of H₂ is volumetric percentage basis. The COY is the brand name for anaerobic chambers from the US company (COY Laboratory Products, United States). We have specified the information in the revised manuscript.

Q23. Line 495 states that 20 mM of each amino acid was used, unless otherwise noted. Why was this concentration used for amino acids? Is there evidence to cite as justification that 20 mM of a singular amino acid in a 100 mM phosphate buffer (pH 7) solution is representative of a plausible icy planetesimal environment?

A23: Thanks for the comment. Unfortunately, a good estimation of amino acid in the icy planetesimal environment has not been reported. The available data from the chondrites are the weight percentages with respect to the whole rock. The reported Gly abundances in various carbonaceous chondrites range from 3.46~770 nmol/g (Table S4). Considering olivine ((Mg,Fe)₂SiO₄) as the starting material, and Fe,Mg-phyllsilicate ((Mg,Fe)₃Si₂O₅(OH)₄) is the major component of secondary mineral after aqueous alteration, the mass of the rock increases by 1.36~2.64 times due to hydration. Supposing that the water/rock ratio ranges from 0.1 to 10, Gly concentration can be estimated to be 0.047~20.328 mM in the aqueous phase. Therefore, we have used 20 mM as a reasonable starting concentration. In the meantime, to evaluate the dependence of the reactivity on Gly concentration, we have conducted the electrolysis with a wide range of concentrations (0.2~20 mM), and the data have been provided in Fig. S14. We have also tested the reactivity

of other amino acids (Ala and Val) at 2 mM concentrations, to clarify the reaction scheme under low concentration conditions. Several tens of mM concentrations have also been applied in other studies of amino acid stability under hydrothermal conditions (e.g., Islam et al., 2003; Kohara, et al., 1997). We believe that a good comparison between our data and those in these studies can be made due to the similar starting concentrations. We cited the related literature in the revised manuscript and added one sentence in Lines 98~99: “The concentration is comparable to other studies where the stability of amino acids was investigated (51-53)”.

Q24. Lines 496-498 state: “Argon gas was purged into the solution (20 ml min⁻¹) for at least 1 hour prior to the electrolysis. After purging, the solutions were sampled as blank. While keeping the Ar purging, a constant potential was applied on the working electrode by using a multi-channel potentiostat (PS-08; Toho Technical Research).”

Presumably, the need to purge the system with Ar was similar for why the catalysts were prepared in a glove box filled with Ar, which was to prevent oxidation by atmospheric oxygen, as the presence of molecular oxygen can alter the composition of the reaction products. If so, how did collection of the blank occur without breaking the seal and allowing oxygen to enter the reaction apparatus? More generally, the reader needs to know the specific steps used to execute the sampling process that was implemented, so that the reader can properly reproduce the work detailed in this manuscript.

A24: Thanks for raising this concern. The sampling port is through the outlet port on the cap. We have revised the Fig. S1 to show the sampling port and described this setup in the caption of Fig. S1 as follows:

The silicon cap has three holes for inserting a gas channeling glass tube, a reference electrode and the gas outlet/sampling port.

Q25. Line 501: an explanation for equation (2), or a citation to refer readers to that might otherwise explain equation (2), would be helpful for readers to better understand the context of equation (2). As a side note, there is no equation (1) in the is manuscript, but the manuscript does include equations (2), (3), and (4). The numbering scheme for equations need to start with (1).

A25: A new citation (ref. 94) was added to guide the readers to understand the equation (2). Sorry for the mistake in the notation of equations. We have corrected the numbering of each equations. Since a new equation was added in the text (refer to the reply to Q34), the numbers for other equations do not change.

Q26 Many of the figures contained graphics that were of poor resolution or challenging to interpret. Examples are listed below:

a. The “light gray dots” in Fig. 2B are too small and too difficult to see. They need to be a brighter or darker color.

A26: we have revised the color to darker color.

Q27. The NMR spectra in Figures 4, S8-S13 are of poor quality and must be improved, particularly to allow the reader to better see the small peaks that were pointed out in these figures.

A27: we have enlarged the NMR spectra in a new figure (Fig. 5) in the revised manuscript to show the peaks clearly. We also made the lines bold or enlarge the figures to make the small peaks clearer in supplementary Fig. S8~S13.

Q28. The data points in Figure 5 are often so tightly clustered together and of similar colors, making it very difficult to follow the data point trends throughout the subfigures. A combination of color and symbol variations are recommended for use in this figure, as opposed to using the same symbol for each data point and just changing the colors.

A28: We agree with the suggestion from the reviewer, and revised the graphics to make them clearer.

Q29 Lines 180-182 indicated that HPLC concentration measurement errors were <2%; however, none of the quantitative data contains uncertainty estimates. Even though the HPLC measurement error was small, it is nonetheless imperative to include measurement errors when displaying quantitative data in figures and tables. In particular, Figures 3, 5-7, S3-S5, S14-S15, S17-S18, and Tables S1-S3 and S5-S8 need uncertainty estimates to accompany the measurement estimates.

A29: The uncertainty estimates have been added to the plots in the related figures and tables. Please be noted that the error is sometimes too small to be clearly seen. Additionally, the error estimation of some organic abundances in carbonaceous chondrites were not reported (table S5 and S6, therefore, we couldn't add any error estimates for those data.

Q30 Figure S16 shows an NMR spectrum of reaction products with notable analytes pointed out. However, the reaction product NMR spectrum is not accompanied with appropriate standard NMR spectra. The standard spectra are necessary for readers to properly evaluate the veracity of the claims of analyte identification in the reaction product NMR spectrum.

A30: We agree with the suggestion from the reviewer, and added ¹H-NMR spectra of

authentic standards in Fig. S8~S13 and Fig. S16.

Q31 The manuscript often refers to GRO 95577 as a CR 2.0, and cites Aponte et al. (2020), MAPS, 55, 2422-2439 (e.g., see Table S4 of the manuscript being reviewed). While it is true that Table 1 of Aponte et al. (2020) states that GRO 95577 is a CR 2.0, but in doing so, the Aponte et al. (2020) article cites the GRO 95577 data from Glavin et al. (2010) MAPS, 45, 1948-1972, and Glavin et al. (2010) states that GRO 95577 is a CR1. Furthermore, the Meteoritical Bulletin Database states that GRO 95577 is a CR1 (<https://www.lpi.usra.edu/meteor/metbull.php?sea=GRO+95577&sfor=names&ants=&nwas=&falls=&valids=&stype=contains&lrec=50&map=ge&browse=&country=All&srt=name&categ=All&mblist=All&rect=&phot=&strewn=&snew=0&pnt=Normal table&code=11305>). Please confirm the meteorite classifications used in the manuscript.

A31: The secondary processes or petrologic histories of carbonaceous chondrites have been well studied, initially with the assignment of integer petrologic types ranging from 1 to 6, where type 3 chondrites experienced essentially no post-accretion secondary alteration. Meteorites with types <3 experienced increasing aqueous alteration, and those with type>3 experienced increasing thermal alteration (Van Schmus et al., 1974, *Geochimica et Cosmochimica Acta* 38:47–64; McSween et al., 1979, *Geochimica et Cosmochimica Acta* 43:1761–1770). More recently, new classification schemes have been developed to better distinguish the degrees of aqueous alteration among CR chondrites, based on petrography and oxygen isotopic composition, ranging from petrologic type 2.0 to 2.8 (Harju and Rubin et al., 2014, *Geochimica et Cosmochimica Acta* 139:267-292). Similar methods have been developed to distinguish the progressive aqueous alteration degrees of CM carbonaceous chondrites by Rubin et al. (*Geochimica et Cosmochimica Acta* 2007, 71(9), 2361-2382).

Since these new classification schemes with numerical sequences well describe and differentiate the degree of aqueous alteration, they have been used in recent studies focusing on the organic distribution variations among chondrites with different aqueous alteration degrees, such as CM chondrites by Glavin et al. (*Meteoritics & Planetary Science*, 2020, 55(9):1979-2006), and CR chondrites by Aponte et al. (*Meteoritics & Planetary Science*, 2020, 55(11):2422-2439).

One of the reasons why Galvin 2010 paper didn't use the new classification scheme for denoting GRO 95577 is likely that the publication of Harju et al. paper (in 2014) is later than Glavin et al. paper (in 2010).

Specifically, GRO 95577 was originally described by Weisberg and Huber (*Meteoritics & Planetary Science*, 2007, 42:1495–1503.) as CR1. In Harju et al. paper, they did systematic petrographic and isotopic analyses. Chondrules in GRO 95577 were completely altered

(mainly to phyllosilicates and oxides), but remain as spheroidal pseudomorphs. The mesostases of these chondrules consist of phyllosilicates that appear light green in plain-polarized transmitted light. Metallic Fe-Ni is very rare and occurs only as kamacite cores surrounded by iron oxide. Oxidation of metallic Fe-Ni proceeds initially from the exterior of the grain inwards as evidenced by the oxide rinds surrounding residual kamacite cores. The GRO 95577 matrix contains elongated patches of iron-oxide framboids. The chondrule/matrix modal abundance ratio of GRO 95577 is similar to that of typical CR chondrites (1.5 ± 0.5). Coarse carbonate occurrences have been observed in GRO 95577. A general correlation between $\Delta^{17}\text{O}$ value and the degree of parent-body aqueous alteration was observed, with higher $\Delta^{17}\text{O}$ values observed in GRO 95577. Ferrous sulfate associated with oxides in opaque assemblages was observed and concluded to be a product by oxidation of dissolved sulfur during aqueous alteration. I would like to guide the reviewer to Harju et al. paper for more detailed information.

Therefore, we prefer to use the newly established classification system to better represent and compare the degree of aqueous alteration among CR chondrites.

Q32 The reference to Fig. 3B in Line 194 is incorrect. This needs to be a reference to Fig. 4B.

A32: Thanks. We have corrected the reference.

Q33 A reference is needed to substantiate the statement made in Lines 263-264.

A33: We have added a reference to substantiate the statement and complemented the discussion.

Q34 Lines 53-54 state the hypothesis of this work: "Here, we hypothesize that low-temperature geo-electrochemical reactions altered amino acids in water-rock differentiated CCs' parent bodies (icy planetesimals) due to pH and redox gradients."

The manuscript did not sufficiently provide evidence that the "pH and redox 'gradients'" component of the hypothesis was empirically evaluated and proven via experiments in this work. The experimental design incorporated the use of a pH 7 phosphate buffer and applied "a constant potential" (Line 497) to the working electrode. The manuscript stated that the electrochemical potential range of -0.6 to -1.0 V that is generated from a pH range of 9-13 and a temperature range of 0-150 degC in the icy planetesimal was simulated. Therefore,

the manuscript is claiming that because a specific electrochemical potential range was simulated, then this means that a pH range of 9-13 was also simulated. This appeared to be the basis for the claim that pH and redox gradients were tested in these electrolysis experiments. However, the electrolysis experiments didn't implement any pH or electrical potential gradients. Instead, the experiments entailed the use of constant electrical potentials applied in a constant pH environment. The manuscript needs to better substantiate why individually testing constant electrical potentials in a constant pH environment is equivalent to pH and redox gradients.

A34: Thanks for raising this concern. To clarify the methodology that a constant potential was used to simulate the equivalent pH and redox gradients, we have added the following sentence in the first paragraph of the Results section.

Lines 110~117:

Here, the half reaction of electrochemical alteration of amino acid (Fig. 1) was simulated using electrolysis under a constant potential. The electrode potential represented the magnitude of the redox gradient (H₂ enriched core versus bicarbonate-buffered mantle) and was calculated by Nernst equation:

$$E_h = \frac{1}{2F} (2RT \ln \alpha_{H^+} - RT \ln \alpha_{H_2} - \Delta_f G^0(H_2)) \quad (1)$$

The electrode potential is equivalent to the reduction potential of H₂/H⁺ redox couple under specific pH and temperature conditions as depicted in Fig. 2A. A more negative electrode potential corresponds to a condition with more alkaline pH or higher temperature.

Q35 The results when using variable concentrations of starting glycine reagent are not intuitively understood and were not sufficiently evaluated:

a. Figure S14A and S18 showed that the amount of methylamine produced from glycine at -0.9 V drops off significantly from 2 mM glycine to 20 mM glycine. This result is true whether NiS is the catalyst or FeS is the catalyst. Why would the methylamine production be so much smaller from 20 mM glycine than 2 mM glycine? Wouldn't having a greater amount of starting material facilitate the production of a greater amount of product? This unexpected result needs to be clarified. Furthermore, this trend is notably contradictory to that observed for glycolate production vs glycine concentrations.

A35: Thanks for this comment. We agree with the reviewer that the negative correlation between methylamine concentration and starting glycine concentration is unexpected. We have added discussion on this result in the main text as below:

Lines 179~183:

Notably, glycolate yield increases upon elevating the starting Gly concentration, suggesting

that Gly molecule was involved in the rate-limiting step of glycolate production pathway. Unexpectedly, methylamine showed increased yield with lowering starting Gly concentration. This result indicated that methylamine generation competed with glycolate generation and was less dependent on the Gly concentration.

RECOMMENDATION

Reject. Explanation for recommendation provided below.

This manuscript has the potential to be of interest to the meteoritic community, particularly the component of this community that actively studies soluble organics. However, the scope of this work comes across as too narrow and specific to be of broad appeal to the greater scientific community. Based on the weaknesses described above, this manuscript does not appear appropriate for publication in a journal with an impact factor of ~15 (i.e., Nature Communications).

There is substantial work that needs to be done to improve this manuscript. However, after improvements have been made, it is possible this work could be suitable for publication in a journal that serves the specific community of interest. Example target journals include Meteoritics & Planetary Science, Geochimica Cosmochimica Acta, or Earth and Planetary Science Letters, to name a few.

Hopefully the authors will take these recommendations under consideration because the basic tenants of the study are interesting and worthy of consideration for publication, but in a different venue than Nature Communications, and after significant improvements have been made to the manuscript.

In addition to the above revisions, we have revised another sentence for correcting the improper use of “membrane” (polyester droplets are membraneless structure):

Lines 331~332: α -hydroxy acids were building blocks of the polyester membrane, serving as a type of proto-compartment for encapsulating and stabilizing biomolecules

Revised to: α -hydroxy acids were building blocks of the polyester droplets, serving as a type of proto-compartment for encapsulating and stabilizing biomolecules

REVIEWERS' COMMENTS

Reviewer #2 (Remarks to the Author):

The authors have satisfactorily addressed all of the reviewer comments, including performing several new experiments to strengthen the manuscript. I am happy to recommend the article for publication and think it will be of broad interest to the origins of life, prebiotic chemistry, and meteoritics communities.

Reviewer #4 (Remarks to the Author):

The manuscript details experimental results of the electrolysis of amino acids under conditions simulating aqueous alteration of carbonaceous chondrites. The authors found that electrolysis under these conditions leads to the formation of alkyl amines and carboxylic acids when carried out in the presence of Fe and Ni sulfides. The manuscript is well written and the SI provides a thorough presentation of the raw data (IC/HPLC chromatograms and NMR spectra) from the experiments, controls, and standards which enables the reader to carefully assess the presented results. Overall, I found the manuscript a pleasure to read and I believe the results will be of broad interest to the astrobiology/origins of life/meteorite/planetary science communities. The trends regarding amino acids, amines, and hydroxy acids in aqueously altered meteorites is well noted and this manuscript provides a strongly plausible explanation for this observation. Moreover, because the work invokes conditions that were likely common on other planetary bodies including the early Earth, Ceres, and Enceladus – the mechanism studied here could have important implications for origins of life and help inform future missions that aim to characterize the organic content of other planetary bodies. As the manuscript provides a testable hypothesis which can be readily explored by the community, it seems highly likely that this work would motivate new studies, both meteoritic and experimental, to determine the possibility of it being true.

See attached document for minor comments with suggested edits.

The manuscript details experimental results of the electrolysis of amino acids under conditions simulating aqueous alteration of carbonaceous chondrites. The authors found that electrolysis under these conditions leads to the formation of alkyl amines and carboxylic acids when carried out in the presence of Fe and Ni sulfides. The manuscript is well written and the SI provides a thorough presentation of the raw data (IC/HPLC chromatograms and NMR spectra) from the experiments, controls, and standards which enables the reader to carefully assess the presented results. Overall, I found the manuscript a pleasure to read and I believe the results will be of broad interest to the astrobiology/origins of life/meteorite/planetary science communities. The trends regarding amino acids, amines, and hydroxy acids in aqueously altered meteorites is well noted and this manuscript provides a strongly plausible explanation for this observation. Moreover, because the work invokes conditions that were likely common on other planetary bodies including the early Earth, Ceres, and Enceladus – the mechanism studied here could have important implications for origins of life and help inform future missions that aim to characterize the organic content of other planetary bodies. As the manuscript provides a testable hypothesis which can be readily explored by the community, it seems highly likely that this work would motivate new studies, both meteoritic and experimental, to determine the possibility of it being true.

It should be noted in the discussion that the experimental conditions, which generates ammonia/amines and alpha keto acids, could then lead to subsequent reductive amination reactions and the reformation of amino acids as shown by Barge et al., 2019. However, because these are strongly reducing conditions reductive amination to reform amino acids may not occur as it has been shown that Fe(II) minerals do not promote reductive amination but rather the formation of alpha hydroxy acids (Barge et al., 2019). Were there any measurements performed on the Fe/Ni sulfides after the experiments? This could help inform the degree to which reductive amination could have occurred following the initial electrolysis of the amino acid.

Minor comments:

Line 514: “In the interface, the amino acids would have been, then, converted into soluble compounds, such as amine (including ammonia), α -hydroxy acids, and monocarboxylic acids (Fig. 9, ②).” Soluble should be changed to derivative or some other word since amino acids are also soluble.

Line 515: “A part of these soluble organics could have been, again, transported..” Change “a part of” to “some of”

Line 540: “Amino acids were typically considered sTable...” Lowercase the “T” in stable.

Line 532: “The surface materials of Ceres would reflect materials contained in the subsurface ocean at the earlier stage of its evolution (92).” To improve readability and soften the certainty of this statement

change to, “The surface materials of Ceres would may reflect materials contained in from the an ancient subsurface ocean at the earlier stage of its evolution (92).”

Line 541: “however, our results suggested...” change “suggested” to “suggests”

Line 542: “...could alter amino acids into various amino- and carbonyl-bearing compounds with small molecular weight (e.g., ammonia, methylamine, ethylamine, formate, acetate)...” change to “...could alter degrade amino acids into lower molecular weight various amines and organic acids amino and carbonyl-bearing compounds with small molecular weight (e.g., ammonia, methylamine, ethylamine, formate, acetate)...”

Line 548: “The plume includes low-mass amine and carbonyls (93), which is also consistent with our results of derivative products of amino acids.” Clarify that you are talking about the Enceladus plume. In addition, as is it is unclear what the main message of this sentence is. It seems that the authors are stating that the amines and carbonyls in the Enceladus plume are from amino acid degradation, but this cannot be proven. Moreover, several sources could contribute to those organics. Therefore, I strongly suggest the authors rewrite the sentence to say “The Enceladus plume contains includes low-mass amine and carbonyls (93), which is also which could have been partially derived from the our results of derivative products of degradation of amino acids via electrolysis as shown in this work.”

Line 554: “Sampling and analyzing their surface materials would clarify the existence of amino acids and mineral catalysts and testify their relations to other compounds.” Instead of “clarify” say “confirm.” This sentence should be rewritten to state how the hypothesis proposed here could be tested by sampling Ceres or Enceladus. What evidence would support the idea that the organics present were derived from amino acid electrolysis?

The figure captions in the SI for Figs. S1-S4, S9-S15, S19, S21 should be kept on the same page as the figure, otherwise it gets confusing. Currently the figure captions are offset so that they are on different pages then the figure they are describing. This could be easily fixed.

Make sure lines 190 and 191 remain on the same page as the figure caption for S22, it looks like extra space was added for no reason. Same with lines 197 and 198 for S23; lines 203 and 204 for Fig. S24

Change “show” to “shows” in lines 202, 203, 208, 209 of the SI.

Reviewer #5 (Remarks to the Author):

In the manuscript entitled "Goelectrochemistry-driven alteration of amino acids to derivative organics in carbonaceous chondrite parent bodies," Li and co-authors presented experimental data on the goelectrochemical alteration of amino acids in the context of icy planetary bodies. This work is novel as it provides some new hypotheses in an underexplored area, with relevance to origins of life, life detection, and potential upcoming missions. The authors initially received three sets of comprehensive review comments and I thought they revised the manuscript appropriately and thoroughly. The addition of the new experiments and clarification of some of their methodology (including the blanks) helped improve the overall manuscript. I was happy to see that the readability of the manuscript was improved through the review and revision process, but I still have a couple of concerns that need to be addressed in terms of word choices, grammar, spelling, and sentence structure. I have provided a document that lists some line edits that addresses some of these concerns. Overall, I think this is a solid manuscript that was drastically improved from its original submitted version and I commend the authors for their detail and diligence to addressing the concerns of the initial reviewers.

Remove) in line 169 after respectively

Line 248 Figure 5 caption, 'evidence' instead of evidences

Line 257: use a – instead of ~ between Fig. S22-S25

Line 258: replace 'from' with 'with' near end of sentence, "starting **with** Asp..."

Line 260: 'indicate' missing an s at the end (indicates)

Line 256: 'suggest' missing an s at the end (suggests)

Line 266: 'investigated in **the** future' or 'investigated in future **experiments**'

Line 270: replace majorly with another adjective, like greatly

Line 271: mostly -> most

Line 275: replace hasn't with haven't (or have not)

Line 299: 'through **the** inductive effect'

Line 313: replace metaphorized with metamorphized

Line 313: replace activities with alteration (end of sentence)

Line 360: typo in the second occurrence of hydroxy (in hydroxy acids), at end of sentence

Line: 'due to variations in the samples'

Line 408: Constrain -> constraining

Line 438 & 441: dominate -> dominant

Line 447: replace with 'radionuclide'

Line 493: errors -> error

Line 494: remove firmly

Line 540: sTable -> stable

General: when reporting value ranges, use – instead of ~

Response to reviewers

Geoelectrochemistry-driven alteration of amino acids to derivative organics in
carbonaceous chondrite parent bodies (NCOMMS-21-49841)

Yamei Li, Norio Kitadai, Yasuhito Sekine, Hiroyuki Kurokawa, Yuko Nakano, and Kristin
Johnson-Finn

We thank the reviewers and editors for providing constructive comments. In the revised version of our manuscript, we addressed all of the points raised by the reviewers. We highlighted the revision in blue in the revised manuscript. Please see below for a point-by-point response to the reviewers' comments.

REVIEWERS' COMMENTS

Reviewer #2 (Remarks to the Author):

The authors have satisfactorily addressed all of the reviewer comments, including performing several new experiments to strengthen the manuscript. I am happy to recommend the article for publication and think it will be of broad interest to the origins of life, prebiotic chemistry, and meteoritics communities.

Reply: We appreciate the positive comments highlighting the potential scientific contribution of our study.

Reviewer #4 (Remarks to the Author):

The manuscript details experimental results of the electrolysis of amino acids under conditions simulating aqueous alteration of carbonaceous chondrites. The authors found that electrolysis under these conditions leads to the formation of alkyl amines and carboxylic acids when carried out in the presence of Fe and Ni sulfides. The manuscript is well written and the SI provides a thorough presentation of the raw data (IC/HPLC chromatograms and NMR spectra) from the experiments, controls, and standards which enables the reader to carefully assess the presented results. Overall, I found the manuscript a pleasure to read and I believe the results will be of broad interest to the astrobiology/origins of life/meteorite/planetary science communities. The trends regarding amino acids, amines, and hydroxy acids in aqueously altered meteorites is well noted and this manuscript provides a strongly plausible explanation for this observation. Moreover, because the work invokes conditions that were likely common on other planetary bodies including the early Earth, Ceres, and

Enceladus – the mechanism studied here could have important implications for origins of life and help inform future missions that aim to characterize the organic content of other planetary bodies. As the manuscript provides a testable hypothesis which can be readily explored by the community, it seems highly likely that this work would motivate new studies, both meteoritic and experimental, to determine the possibility of it being true.

Reply: We appreciate the positive comments highlighting the potential scientific contribution of our study.

See attached document for minor comments with suggested edits.

Q1: It should be noted in the discussion that the experimental conditions, which generates ammonia/amines and alpha keto acids, could then lead to subsequent reductive amination reactions and the reformation of amino acids as shown by Barge et al., 2019. However, because these are strongly reducing conditions reductive amination to reform amino acids may not occur as it has been shown that Fe(II) minerals do not promote reductive amination but rather the formation of alpha hydroxy acids (Barge et al., 2019). Were there any measurements performed on the Fe/Ni sulfides after the experiments? This could help inform the degree to which reductive amination could have occurred following the initial electrolysis of the amino acid.

A1: Thanks for the comment. Under the present experimental conditions, electrolysis of amino acids does not generate alpha keto acids, therefore, there will be no subsequent reductive amination reaction.

Q2 Minor comments:

Line 514: “In the interface, the amino acids would have been, then, converted into soluble compounds, such as amine (including ammonia), α -hydroxy acids, and monocarboxylic acids (Fig. 9, ②).” Soluble should be changed to derivative or some other word since amino acids are also soluble.

Line 515: “A part of these soluble organics could have been, again, transported..” Change “a part of” to “some of”

Line 540: “Amino acids were typically considered sTable...” Lowercase the “T” in stable.

Line 532: “The surface materials of Ceres would reflect materials contained in the subsurface ocean at the earlier stage of its evolution (92).” To improve readability and soften the certainty of this statement change to, “The surface materials of Ceres would may reflect materials contained in from the an ancient subsurface ocean at the earlier stage of its evolution (92).”

Line 541: “however, our results suggested...” change “suggested” to “suggests”

Line 542: “...could alter amino acids into various amino- and carbonyl-bearing compounds with small molecular weight (e.g., ammonia, methylamine, ethylamine, formate, acetate)...”

change to "...could alter degrade amino acids into lower molecular weight various amines and organic acids amino- and carbonyl-bearing compounds with small molecular weight (e.g., ammonia, methylamine, ethylamine, formate, acetate)..."

Line 548: "The plume includes low-mass amine and carbonyls (93), which is also consistent with our results of derivative products of amino acids." Clarify that you are talking about the Enceladus plume. In addition, as is it is unclear what the main message of this sentence is. It seems that the authors are stating that the amines and carbonyls in the Enceladus plume are from amino acid degradation, but this cannot be proven. Moreover, several sources could contribute to those organics. Therefore, I strongly suggest the authors rewrite the sentence to say "The Enceladus plume contains includes low-mass amine and carbonyls (93), which is also which could have been partially derived from the our results of derivative products of degradation of amino acids via electrolysis as shown in this work."

Line 554: "Sampling and analyzing their surface materials would clarify the existence of amino acids and mineral catalysts and testify their relations to other compounds." Instead of "clarify" say "confirm." This sentence should be rewritten to state how the hypothesis proposed here could be tested by sampling Ceres or Enceladus. What evidence would support the idea that the organics present were derived from amino acid electrolysis?

The figure captions in the SI for Figs. S1-S4, S9-S15, S19, S21 should be kept on the same page as the figure, otherwise it gets confusing. Currently the figure captions are offset so that they are on different pages than the figure they are describing. This could be easily fixed.

Make sure lines 190 and 191 remain on the same page as the figure caption for S22, it looks like extra space was added for no reason. Same with lines 197 and 198 for S23; lines 203 and 204 for Fig. S24

Change "show" to "shows" in lines 202, 203, 208, 209 of the SI

A2: Thanks for the careful reviewing on English language issues. We appreciate these comments and have revised accordingly.

The sentence in line 554 (previous version manuscript) has been rewritten as follows:

Lines 456~460 (new version manuscript): Sampling and analyzing their surface materials would confirm the existence of amino acids, derivative compounds (e.g., amines, hydroxy acids), and mineral catalysts. Supposing a relative enrichment of derivative compounds with respect to their amino acid analogs is detected in a sulfide-rich sample, this may indicate electrolysis of amino acids has operated and affected the organic distribution on these icy bodies.

Reviewer #5 (Remarks to the Author):

Q3: In the manuscript entitled “Geoelectrochemistry-driven alteration of amino acids to derivative organics in carbonaceous chondrite parent bodies,” Li and co-authors presented experimental data on the geoelectrochemical alteration of amino acids in the context of icy planetary bodies. This work is novel as it provides some new hypotheses in an underexplored area, with relevance to origins of life, life detection, and potential upcoming missions. The authors initially received three sets of comprehensive review comments and I thought they revised the manuscript appropriately and thoroughly. The addition of the new experiments and clarification of some of their methodology (including the blanks) helped improve the overall manuscript. I was happy to see that the readability of the manuscript was improved through the review and revision process, but I still have a couple of concerns that need to be addressed in terms of word choices, grammar, spelling, and sentence structure. I have provided a document that lists some line edits that addresses some of these concerns. Overall, I think this is a solid manuscript that was drastically improved from its original submitted version and I commend the authors for their detail and diligence to addressing the concerns of the initial reviewers.

A3: We appreciate the positive comments highlighting the potential scientific contribution of our study. Thanks for the careful reviewing on English language issues. We have revised these problems accordingly and highlighted in blue in the main manuscript.